# Ratiometric sensing of BiP-client versus BiP levels by the unfolded protein response determines its signaling amplitude

Anush Bakunts[1†], Andrea Orsi[1,2†], Milena Vitale[1,2†], Angela Cattaneo[3], Federica Lari[1], Laura Tadè[1], Roberto Sitia[1,2], Andrea Raimondi[4], Angela Bachi[3], Eelco van Anken[1,2]*

[1]Division of Genetics and Cell Biology, San Raffaele Scientific Institute, Milan, Italy; [2]Università Vita-Salute San Raffaele, Milan, Italy; [3]IFOM, FIRC Institute of Molecular Oncology, Milan, Italy; [4]Experimental Imaging Center, San Raffaele Scientific Institute, Milan, Italy

**Abstract** Insufficient folding capacity of the endoplasmic reticulum (ER) activates the unfolded protein response (UPR) to restore homeostasis. Yet, how the UPR achieves ER homeostatic readjustment is poorly investigated, as in most studies the ER stress that is elicited cannot be overcome. Here we show that a proteostatic insult, provoked by persistent expression of the secretory heavy chain of immunoglobulin M ($\mu_s$), is well-tolerated in HeLa cells. Upon $\mu_s$ expression, its levels temporarily eclipse those of the ER chaperone BiP, leading to acute, full-geared UPR activation. Once BiP is in excess again, the UPR transitions to chronic, submaximal activation, indicating that the UPR senses ER stress in a ratiometric fashion. In this process, the ER expands about three-fold and becomes dominated by BiP. As the UPR is essential for successful ER homeostatic readjustment in the HeLa-$\mu_s$ model, it provides an ideal system for dissecting the intricacies of how the UPR evaluates and alleviates ER stress.

DOI: https://doi.org/10.7554/eLife.27518.001

*For correspondence:
evananken@mac.com

[†]These authors contributed equally to this work

**Competing interests:** The authors declare that no competing interests exist.

## Introduction

Key to survival is that organisms adapt their behavior—or on the cellular level, their molecular machineries—according to need. An example is the ability of the endoplasmic reticulum (ER) to accomodate fluctuations in the load of client proteins that fold and assemble in this organelle before being dispatched to travel further along the secretory pathway (*Walter and Ron, 2011*). A vast array of ER resident chaperones and folding factors assist the maturation of client proteins (*van Anken and Braakman, 2005*). BiP is an ER resident HSP70 family chaperone, whose function is regulated by at least seven different HSP40 co-chaperones, named ERdj1–7, and two nucleotide exchange factors (*Melnyk et al., 2015*), as well as through reversible AMPylation (*Preissler et al., 2015*). The ER also hosts an HSP90 family member, GRP94, for which CNPY family proteins may be co-chaperones (*Anelli and van Anken, 2013*), and several peptidylprolyl isomerases (PPIases), such as cyclophilin B and a few FKBP family members. Special to the ER are the protein disulfide isomerases (PDI) that assist disulfide bond formation, and the lectin chaperones, such as calnexin (CNX) and calreticulin (CRT), that associate with client proteins through binding their N-glycans (*van Anken and Braakman, 2005*).

The chaperones and folding factors retain folding and assembly intermediates in the ER in order to minimize the secretion or deployment throughout the endomembrane system of

immature client proteins (*Ellgaard and Helenius, 2003*). A consequence of client retention is that when the protein folding machinery is not adequate to facilitate maturation of clients, they accumulate in the ER, which causes stress and activates the unfolded protein response (UPR) (*Walter and Ron, 2011*). The circuitry of the UPR has been mapped to impressive detail: the three well-studied branches of the UPR are initiated by the UPR transducers IRE1$\alpha$, PERK and ATF6$\alpha$, each of which has an ER stress-sensing domain in the ER lumen and a UPR signaling effector domain in the cytosol. Activated IRE1$\alpha$ and PERK are kinases that trans-autophosphorylate. This step triggers IRE1$\alpha$ to assume endonuclease activity, such that it removes an intron from *XBP1* mRNA. Upon its religation, the spliced mRNA encodes the XBP1 transcription factor (*Yoshida et al., 2001*; *Calfon et al., 2002*). Activated PERK transiently attenuates protein synthesis through phosphorylation of the translation initiation factor eIF2$\alpha$ (*Harding et al., 1999*). At the same time, eIF2$\alpha$ phosphorylation favors the expression of a few transcripts, in particular ATF4, a transcription factor that activates further downstream effectors, such as CHOP (*Walter and Ron, 2011*). The third UPR branch is activated by ATF6$\alpha$, which undergoes regulated intramembrane proteolysis in the Golgi and thus a transcriptionally active N-terminal portion of 50 kDa is liberated that acts as a transcription factor (*Ye et al., 2000*). The UPR transcription factors jointly initiate genetic programs that drive the expression of all of the components that are necessary to expand the ER, including the chaperones and enzymes for membrane synthesis (*Walter and Ron, 2011*). In fact, overexpression of for instance XBP1 alone leads to ER expansion even in the absence of any perturbation of the ER client protein folding and assembly process (*Sriburi et al., 2004*). Altogether, the UPR homeostatically readjusts the ER folding machinery by expanding the organelle according to need, and regulates cell fate decisions depending on the severity of ER stress (*Walter and Ron, 2011*).

To date, most studies on the UPR circuitry have focused on the signaling pathways themselves, and little is known about how the UPR evaluates the severity of ER stress and the success of the homeostatic readjustment of the ER. Here, we show that the widely used strategy of employing ER stress-eliciting drugs obscures how ER homeostatic readjustment may be achieved, and instead, we present a HeLa cell model that allows us to evaluate just that. By inducible overexpression of orphan immunoglobulin M (IgM) secretory heavy chain ($\mu_s$), we provoke a full-blown UPR, which is essential for the cells to cope with the proteostatic insult. As $\mu_s$ accumulates in the ER, it transiently eclipses BiP levels, at which point the UPR output is strongest. UPR-driven upregulation then allows BiP to reach levels that exceed $\mu_s$ levels again, while the ER expands in the process. The activation of the UPR is maximal only when there is a relative shortage of BiP, whereas it subsides to chronic, submaximal output levels when ER homeostatic readjustment is achieved. Co-expression of Ig light chain ($\lambda$) instead leads to productive IgM secretion, such that BiP is not sequestered by $\mu_s$, the UPR is not activated and the ER does not expand. Thus, the UPR senses the levels of client proteins that sequester BiP versus those of BiP itself in a ratiometric fashion, which determines the amplitude of the response.

## Results

### Cytotoxicity of UPR-eliciting drugs

Drugs that are typically used to study the UPR include tunicamycin (Tm), which prevents the addition of N-glycans to nascent ER client proteins, dithiothreitol (DTT), which impedes disulfide bond formation, and thapsigargin (Tg), which depletes $Ca^{2+}$ from the ER lumen. Their immediate effect is a general collapse of productive protein folding in the ER, and these drugs therefore activate the UPR (*Walter and Ron, 2011*). In the longer run, however, these drugs likely have pleiotropic effects. Evidently, non-productive folding in the ER causes ER client proteins to be retained in the ER (*Ellgaard and Helenius, 2003*), and thus they no longer reach their destination, be it anywhere throughout the endomembrane system of the cell or extracellular. As a result—depending on their half-life—these proteins will be depleted at the site where they have to exert their functions, which ultimately may cause a plethora of perturbations to the homeostasis of the cell.

In order to unmask pleiotropic effects of UPR-eliciting drugs, taking Tm as an example, we serially diluted the drug starting from a concentration (5 µg/ml), which is widely used to study the effects of ER stress and UPR activation, and monitored cell survival after 7 days by a cell growth/colony formation assay. At this high concentration, a full-blown UPR was triggered, as is evident from the maximal

extent of *XBP1* mRNA splicing in HeLa cells, that is, the appearance of a band of higher mobility, corresponding to the RT-PCR product of the $XBP1^S$ transcript from which the intron has been removed (*Figure 1A*). Yet, there was no cell survival after 7 days (*Figure 1B*), not even when the cells were treated with Tm only on the first day for 4 hr. At a 100-fold dilution of the 'standard' concentration, Tm induced the splicing of *XBP1* mRNA to submaximal levels, and only transiently so (*Figure 1A*), but still provoked a substantial cell growth defect (*Figure 1B,C*). Even at a further two-fold dilution of Tm to 25 ng/ml, when no *XBP1* mRNA splicing was detected (*Figure 1A*), there was some loss of cell growth (*Figure 1B,C*). These findings thus reveal that an overall impairment of ER client protein folding has cytotoxic consequences aside from the activation of the UPR. Indeed,

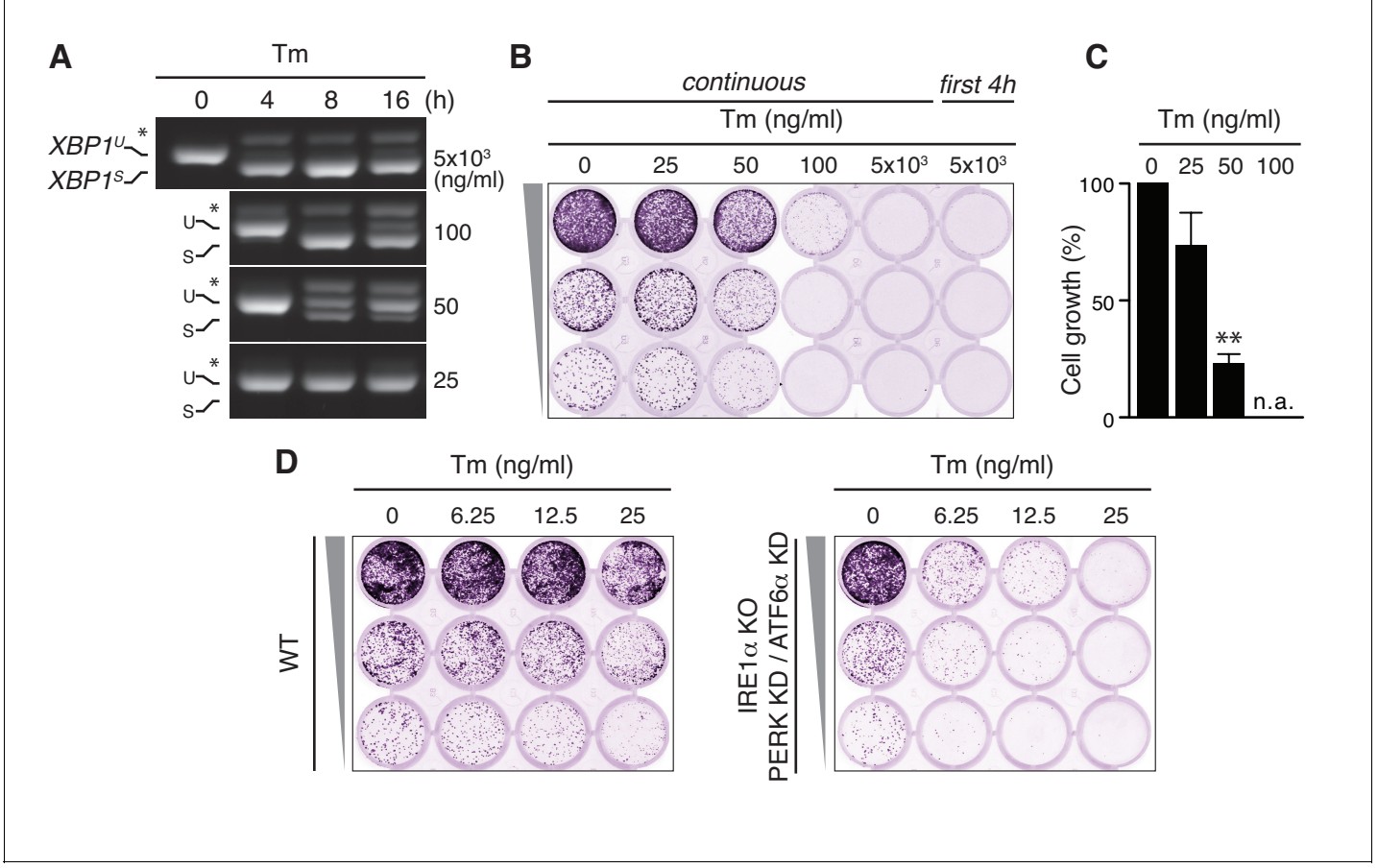

**Figure 1.** Tunicamycin has pleiotropic cytotoxic effects. (A) HeLa S3 cells were treated with Tm at various concentrations and for various durations, as indicated, before mRNA was isolated from cells for RT-PCR analysis with oligos specific for XBP1. PCR fragments corresponding to spliced ($XBP^S$) and unspliced ($XBP^U$) XBP1 were separated on gel. A hybrid product that is formed during the PCR reaction (*Shang and Lehrman, 2004*) is denoted by an asterisk. The ratio of $(XBP^S)/(XBP^S +XBP^U)$ is indicative of the extent of IRE1α activation. (B) Cells as in (A) were seeded upon 1:5 serial dilution into 24-well plates and treated continuously or only for the first 4 hr upon seeding with various concentrations of Tm, as indicated. After 7 days of growth, cells were fixed and stained with crystal violet. (C) Quantitation of the crystal violet staining shown in (B) as a measure of cell growth, except for conditions that fully abrogated growth; n.a. = not assessed. Staining of untreated cells was set at 100%, and values are shown in a bar graph. Mean and s.e.m. are shown; n = 3. Statistical significance in a one-sample *t*-test of differences in crystal violet staining compared to untreated samples in (B) and replicate experiments was determined as a proxy for growth (**p≤0.01). (D) Cell growth assay as in (B) of HeLa S3 cells in which UPR transducers were deleted (IRE1α KO) or silenced (PERK KD and ATF6α KD) or not (WT), treated with Tm at various concentrations, as indicated.

DOI: https://doi.org/10.7554/eLife.27518.002

The following source data and figure supplement are available for figure 1:

**Source data 1.** Data and calculations that were used to generate the bar graph in *Figure 1C*.

DOI: https://doi.org/10.7554/eLife.27518.004

**Figure supplement 1.** Efficiency of ATF6α and PERK silencing.

DOI: https://doi.org/10.7554/eLife.27518.003

Tm readily causes morphological aberrations of the ER at these low concentrations (*Rutkowski et al., 2006*). Strikingly, upon deletion of IRE1α by CRISPR/Cas9 and concomitant ablation of PERK and ATF6α by siRNA-mediated silencing (with good efficiency; *Figure 1—figure supplement 1*), cell growth was hardly impaired. The cytotoxic effects of Tm, however, became apparent in these cells at even lower concentrations than in cells with a functional UPR (*Figure 1D*), which implies not only that the UPR is predominantly cyto-protective but also that UPR-mediated pro-apoptotic pathways cannot play a decisive role in cell fate upon Tm-driven ER stress. Thus, the use of ER stress-eliciting drugs inevitably obscures how ER homeostatic readjustment can be achieved.

## Proteostatic induction of the UPR as a model to assess ER homeostatic readjustment

We reasoned that pleiotropic effects of UPR-eliciting drugs likely obscure various important aspects of how cells cope with ER stress. We therefore sought to circumvent the shortcomings of using ER stress-eliciting drugs, and we hypothesized that overexpression of an ER client protein would be a better tool to allow us to appreciate UPR-driven adaptations to ER stress. The impressive ER expansion during B cell differentiation (*Wiest et al., 1990*; *van Anken et al., 2003*) inspired us to choose the bulk product of plasma cells, secretory antibody, as an ideal candidate to serve as a proteostatic insult that would drive the UPR. More specifically, we decided to employ the secretory heavy chain of IgM, $\mu_s$, because $\mu_s$ expression in bulk coincides with elaborate ER expansion (*Sitia et al., 1987*) and UPR activation (*Reimold et al., 2001*) during plasma cell differentiation.

The precise role of the UPR in ER expansion during plasma cell development is, however, difficult to assess. B cells anticipate that they will secrete IgM in bulk once they differentiate into plasma cells (*van Anken et al., 2003*), such that ER expansion is also driven by developmental cues (*Iwakoshi et al., 2003*). Accordingly, ER expansion occurs in differentiating B cells even when $\mu_s$ is ablated (*Hu et al., 2009*). To assess the effect on cellular homeostasis solely of sudden $\mu_s$ overexpression, we decided to create a HeLa cell model that allowed inducible expression of murine $\mu_s$, for which we exploited the GeneSwitch system (Thermo Fischer Scientific). By subsequent lentiviral delivery and cloning of integrant cells, we introduced the hybrid nuclear receptor that is activated by the synthetic steroid mifepristone (Mif), yielding HeLa-MifON cells. The $\mu_s$ transgene under the control of the promoter element responsive to that hybrid nuclear receptor was then integrated into the HeLa-MifON cells, yielding HeLa-$\mu_s$ cells.

Expression of $\mu_s$ in HeLa-$\mu_s$ cells in the absence of Mif was undetectable by immunoblotting (*Figure 2A*) or immunofluorescence (*Figure 2B*). The GeneSwitch system is amplified through an autoregulatory feedback loop, whereby the Mif-bound hybrid nuclear receptor upregulates expression of its own gene. This feature likely means that the titration of $\mu_s$ expression levels by varying Mif concentration was not easily reproducible, and that the Mif-driven inducibility was instead an OFF/ON effect (data not shown). A proxy for tunable expression of $\mu_s$ was offered instead by comparing different HeLa-$\mu_s$ clones, which—due to differences in genomic locus or copy number of integrations—each expressed $\mu_s$ at a different level upon induction with Mif, as shown for three individual clones (*Figure 2A*). As illustrated by immunofluorescence of $\mu_s$ (*Figure 2B*), expression was remarkably uniform among cells because they were clonal. Moreover, $\mu_s$ is retained in the ER lumen, as shown by its strict co-localization with the ER marker CRT (*Figure 2C*).

The autoregulatory feedback loop of the GeneSwitch system has the advantage that it allows highly abundant expression of the transgene. Indeed, in the most highly expressing HeLa-$\mu_s$ clone, $\mu_s$ reached intracellular levels that outmatched those in model plasma cells: I.29$\mu^+$ lymphomas that were stimulated with lipopolysaccharide (LPS) for 3 days (*Figure 2D*). Synthesis of $\mu_s$ reached a plateau after 1–2 days of consistently high levels, as assessed by radiolabeling (*Figure 2E,F*). Thus, the proteostatic UPR stimulus is permanent, which implies that the cells must undergo homeostatic readjustment to meet the increased demand on the ER folding machinery.

Using a series of standard assays, we then demonstrated that $\mu_s$ triggered a full-blown UPR. Upon 16 hr of induction of $\mu_s$ expression, splicing of *XBP1* mRNA, phosphorylation of PERK (appearance of a band of slightly lower mobility), expression of PERK's downstream effector CHOP, cleavage of the precursor ATF6α-p90 and the concomitant release of its cytosolic portion p50 were all at levels comparable to those triggered by a conventional Tm- or Tg-elicited UPR in these cells (*Figure 2A*). Treatment of the parental HeLa-MifON cells with a high dose of Mif (10 nM) did not

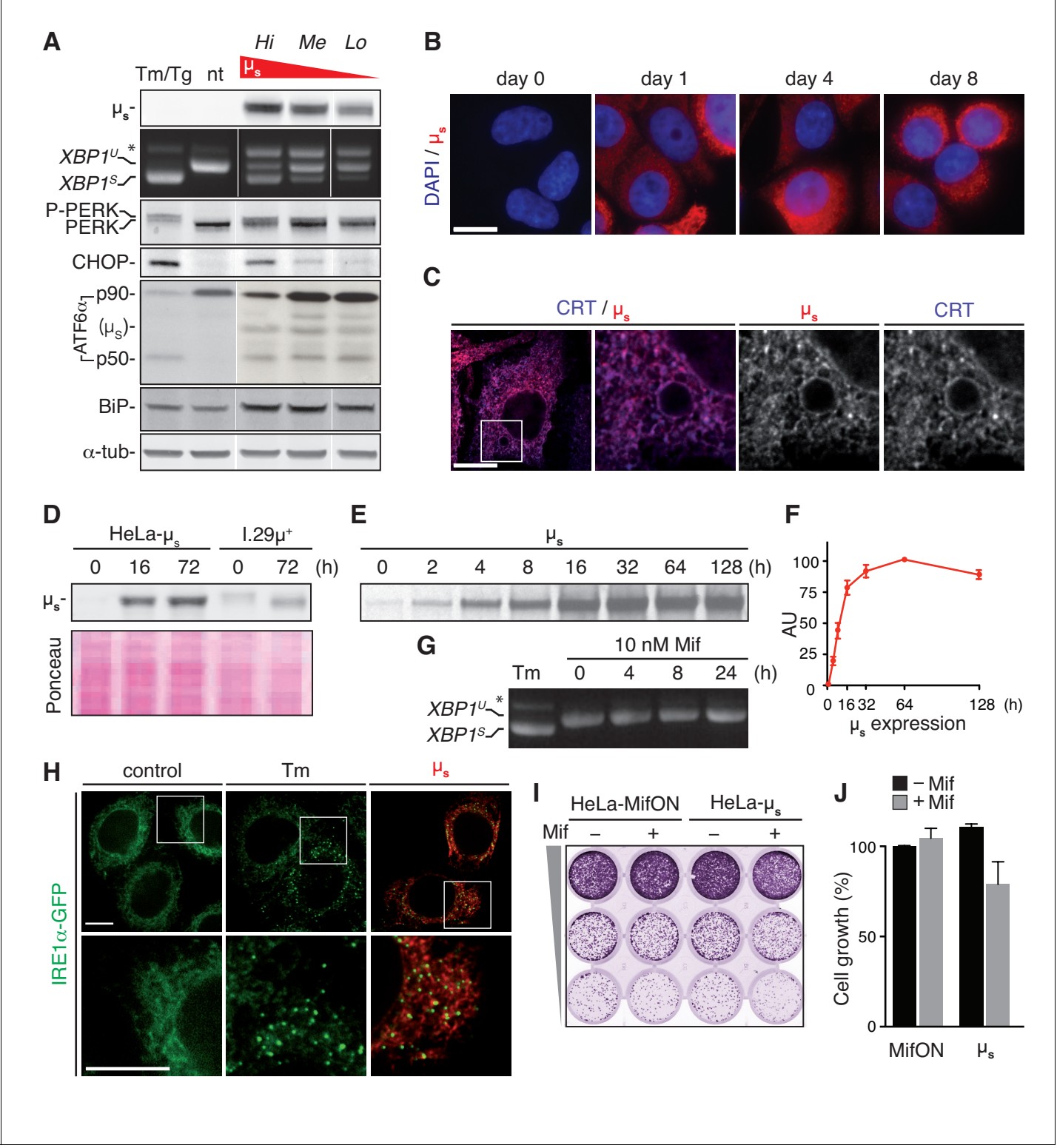

**Figure 2.** A model for proteostatically induced ER stress. (A–F,H–J) Expression of $\mu_s$ in HeLa-$\mu_s$ cells was induced with 0.5 nM Mif. (A,G,H) The UPR was pharmacologically induced as a reference for 4 hr with 5 µg/ml Tm, or—for analysis of ATF6α in (A) only—for 1.5 hr with 300 nM Tg (since Tm leads to deglycosylation of ATF6α). (A,D) Immunoblotting revealed levels of $\mu_s$, CHOP, BiP and α-tubulin. A shift to a lower mobility phosphorylated form, P-PERK, and the appearance of CHOP revealed activation of PERK. The release of the p50 cleavage product from the p90 precursor revealed activation of ATF6α. Cross-reaction of the secondary antibody against anti-ATF6α with $\mu_s$ is denoted. (A,G) Splicing of *XBP1* was assessed as in ***Figure 1A***. (A) Three different clones of HeLa-$\mu_s$ cells with decreasing $\mu_s$ expression levels: high (Hi), medium (Me), or low (Lo) were induced for 16 hr. Non-treated (nt)

*Figure 2 continued on next page*

*Figure 2 continued*

or Tm-treated Hi HeLa-$\mu_s$ served as references. (**B,C,H**) Immunofluorescence revealed $\mu_s$ (red) and CRT (blue), which marks the ER (**C**) at the indicated times before or after induction (**B**) or after 8 hr induction (**C,H**). Nuclei were stained with DAPI (blue) (**B**). (**C,H**) The area that is boxed is shown by 3.5-fold magnification; scale bars represent 10 µm. (**D**) Samples were derived from equal numbers ($7 \times 10^4$) of HeLa-$\mu_s$ or I.29$\mu^+$ lymphomas induced with Mif or stimulated with 20 µg/ml lipopolysaccharide (LPS), respectively, for the indicated times. Ponceau staining of the blot serves as a loading control. (**E**) HeLa-$\mu_s$ cells were pulse labeled with $^{35}$S labeled methionine and cysteine for 10 min at the indicated times after induction. Immunoprecipitated $\mu_s$ was resolved by gel electrophoresis (**F**). Levels of radio-labeled $\mu_s$ in (**E**) were quantified by phosphor imaging. The maximal signal for $\mu_s$ (at 64 hr) was set at 100. (**G**) HeLa-MifON cells were treated with a high dose of Mif (10 nM) for the indicated times. (**H**) In HeLa-$\mu_s$ cells in which IRE1$\alpha$ was replaced with doxycycline (Dox)-inducible IRE1$\alpha$-GFP (green), IRE1$\alpha$-GFP expression was tuned with 10 nM Dox to levels that allowed satisfactory detection of IRE1$\alpha$-GFP by fluorescence microscopy. (**I**) Cell growth assay as in *Figure 1B* of HeLa-MifON and HeLa-$\mu_s$ induced continuously with Mif or not. (**J**) Quantitation of (**I**), performed as in *Figure 1C*. Mean and s.e.m. are shown, n = 2. There is no statistical significance in a one-sample *t*-test of differences in growth between conditions.

DOI: https://doi.org/10.7554/eLife.27518.005

The following source data and figure supplement are available for figure 2:

**Source data 1.** Data and calculations that were used to generate the bar graphs in *Figures 2F and J*.

DOI: https://doi.org/10.7554/eLife.27518.007

**Figure supplement 1.** Functional reconstitution of IRE1$\alpha$-KO cells with IRE1$\alpha$-GFP.

DOI: https://doi.org/10.7554/eLife.27518.006

provoke *XBP1* mRNA splicing (*Figure 2G*). Thus, the UPR induction in Mif-treated HeLa-$\mu_s$ was the result of $\mu_s$ expression. Moreover, UPR activation upon $\mu_s$ induction was dose-dependent, as the extent of *XBP1* mRNA splicing or CHOP expression was commensurate with the levels of $\mu_s$ that were expressed by the different HeLa-$\mu_s$ clones (*Figure 2A*).

Another hallmark of full-blown UPR activation is the clustering of IRE1$\alpha$ (*Li et al., 2010*). We deleted IRE1$\alpha$ by CRISPR/Cas9 and reconstituted it with GFP-tagged murine IRE1$\alpha$ (IRE1$\alpha$-GFP) under the control of a tight Tet-responsive element (Clontech) by lentiviral transduction. The GFP-tag was introduced such that it did not interfere with function (*Figure 2—figure supplement 1*), as has been shown before for human IRE1$\alpha$ (*Li et al., 2010*) or yeast Ire1p (*Aragón et al., 2009*). With doxycycline (Dox), we tuned IRE1$\alpha$-GFP expression to near-endogenous levels that allowed visualization by fluorescent microscopy and found that IRE1$\alpha$-GFP clustered not only upon Tm treatment but also upon $\mu_s$ expression (*Figure 2H*). Altogether, we concluded that $\mu_s$ expression drove the activation of all branches of the UPR that we tested, in a similar manner and to a similar extent as when they were triggered by ER stress-eliciting drugs.

Strikingly, in spite of the expression of $\mu_s$ in bulk, the cell growth of the HeLa-$\mu_s$ cells almost paralleled that of the Mif-treated parental MifON cells (*Figure 2I,J*). Thus, the full-blown UPR that was elicited by $\mu_s$ expression was well tolerated by the cells, unlike prolonged treatment with Tm at concentrations that triggered a full-blown UPR (*Figure 1*). In all, we concluded that the HeLa-$\mu_s$ cells offer a suitable model to assess ER homeostatic readjustment and the role of the UPR in this process, as the cells apparently cope well with the proteostatic insult that is unabated once $\mu_s$ expression is induced.

## UPR activation correlates with accumulation of ER load rather than with the secretory burden

By expressing $\mu_s$, we burden the cells with a Sisyphean task because, in absence of immunoglobulin light chain $\lambda$, no IgM can be produced, a situation that mimics the storage of mutant ER client proteins in disease. Inspired by B cell to plasma cell differentiation, we argued that $\lambda$ in contrast to $\mu_s$ would likely provide a poor if any proteostatic insult to the cells, as it is expressed in B lymphocytes when they still are quiescent, and—by definition—minimally stressed. Moreover, ER retention of $\lambda$ is far less stringent than that of $\mu_s$, such that $\lambda$ can be secreted (*Hendershot and Sitia, 2005*).

To assess whether it is the folding load of client proteins passing through the ER or their accumulation that determines the amplitude of UPR signaling, we created clones that co-express the two IgM subunits and selected clones based on stoichiometric differences in $\lambda$ and $\mu_s$ expression. As expected, when $\lambda$ and $\mu_s$ were co-expressed, antibody assembly took place, and we monitored this by immunoblotting of lysates and of culture media separated by reducing (*Figure 3A*) or non-reducing (*Figure 3B*) gel electrophoresis. The clone in which $\lambda$ was in excess allowed successful

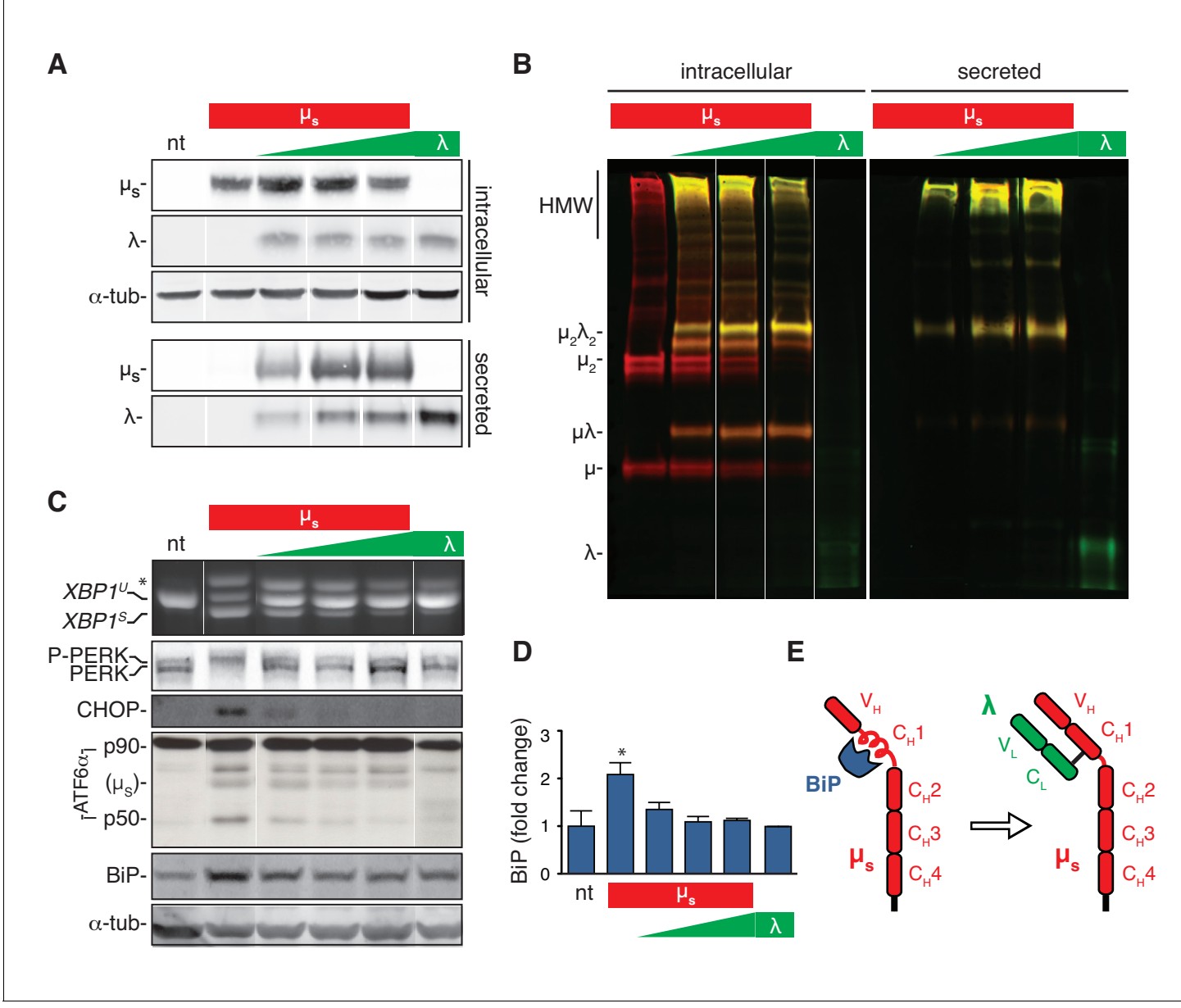

**Figure 3.** The accumulation of clients in the ER rather than the flux of secretory load drives the UPR. (A–D) HeLa clones constructed to express λ inducibly, either alone (λ) or in conjunction with μ$_s$ (μ$_s$ + λ) at different stoichiometries, ranging from μ$_s$ being in excess to λ being in excess, all under control of Mif, were induced with 0.5 nM in parallel to the HeLa-μ$_s$ cells for 16 hr to express IgM subunits. Non-treated (nt) HeLa-μ$_s$ cells served as a reference. Supernatants (secreted) and cell lysates (intracellular) were separated by reducing (A) or non-reducing (B) gel electrophoresis. Levels of μ$_s$, λ (A), CHOP, BiP (C) and α-tubulin (A,C), as well as activation of the UPR pathways (C) were assessed as in *Figure 2A*. (B) Immunoblotting of μ$_s$ (red) and λ (green) reveals disulfide linked assembly intermediates, as indicated: μλ, μ$_2$, μ$_2$λ$_2$, and high molecular weight (HMW) polymeric assemblies of μ$_s$ and λ. Simultaneous immunodetection of both IgM subunits resulted in yellow signal. (D) Quantitation of BiP levels shown in (C) and replicate experiments. Mean and s.e.m. are shown, n = 3. Statistical significance in a one-sample *t*-test of differences in BiP levels is indicated (*p≤0.05). (E) Schematic of BiP associating with the C$_H$1 domain of μ$_s$ until it is displaced by λ (if available).

DOI: https://doi.org/10.7554/eLife.27518.008

The following source data is available for figure 3:

**Source data 1.** Data and calculations that were used to generate the bar graph in *Figure 3D*.
DOI: https://doi.org/10.7554/eLife.27518.009

assembly into polymeric antibodies that were efficiently secreted. In clones where $\mu_s$ was in excess, assembly and secretion were less efficient, leading to intracellular accumulation of disulfide-linked species that are characteristic intermediates of the IgM assembly pathway (*Hendershot and Sitia, 2005*). As anticipated, HeLa cells inducibly expressing λ—at similarly high levels as reached for $\mu_s$—did not activate the UPR (*Figure 3C*). Strikingly, the $\mu_s$-provoked UPR was also almost fully mitigated when λ was co-expressed in excess, while the other clones showed intermediate UPR activation commensurate with an excess of $\mu_s$ (or lack of λ).

As the levels of ER client proteins (the sum of intracellular λ and $\mu_s$) were comparable in the different clones, we concluded that the flux of proteins entering the ER is no key determinant for UPR activation, but rather that the nature of the proteostatic insult determines the extent of UPR activation, with λ being a very poor UPR activator, and $\mu_s$ being a particularly powerful one, provided it does not team up with λ. Of note is that levels of BiP increased markedly when $\mu_s$ was expressed alone but far less so or not at all if λ was present (*Figure 3D*). Indeed, it has been established in plasma cells that BiP binds to $\mu_s$ until it is displaced by λ (*Bole et al., 1986*; *Figure 3E*).

## UPR signaling is commensurate with the extent of BiP being eclipsed by the ER folding load

As the cells that inducibly overexpress $\mu_s$ showed maximal UPR activation with no major negative impact on cell growth—unlike cells treated with UPR-eliciting drugs—we surmised that the $\mu_s$-expressing cells underwent successful homeostatic readjustment of the ER. To appreciate the timing of the response to the proteostatic insult and the process of ER homeostatic readjustment with a more detailed temporal resolution, we undertook further analysis by immunoblotting of the upregulation of $\mu_s$, BiP and of another ER resident chaperone, PDI, with time (*Figure 4A*).

We found that levels of BiP only noticeably increased from 12 hr onwards (and PDI even later), whereas $\mu_s$ built up steadily in the first 12 hr upon induction. Indeed, using recombinant BiP and commercially obtained pure IgM (of which ~70% of the protein mass is $\mu_s$) as standards, we assessed the absolute quantities in cell lysates of BiP and $\mu_s$ upon induction of $\mu_s$ expression. BiP levels increased ~12.5 fold. On the basis of cell counting, determination of the protein weight of the samples and the known molecular weight of BiP of ~70 kDa, we estimated that BiP levels rise from ~$2\times10^7$ to ~$2\times10^8$ copies per cell. Importantly, from the absolute quantitation, we deduced that BiP levels were in excess of $\mu_s$ once the cells had adapted to the proteostatic insult (with $\mu_s$ reaching an estimated ~$1.5$–$1.8\times10^8$ copies per cell). Early upon the onset of $\mu_s$ expression and before BiP induction was fully underway, however, $\mu_s$ levels were transiently at a 1:1 stoichiometry with those of BiP, or possibly, $\mu_s$ levels even slightly exceeded BiP levels (*Figure 4B*).

Following the activation of the UPR over time upon onset of $\mu_s$ expression, we found that all three UPR branches transiently reached an output that was maximal, that is at least on a par with the output that was induced by a conventional drug-elicited UPR (*Figure 4C*; *Figure 4—figure supplements 1–3*). Levels of cleaved ATF6α (p50) reached a maximum at ~6 hr after the onset of $\mu_s$ expression before it subsided to submaximal levels. The full-length precursor ATF6α (p90) was in large part depleted after this intial burst of ATF6α (p50) release (*Figure 4C*), such that the determining factor for the amplitude of signaling through the ATF6α branch was inevitably, the de novo synthesis of p90 rather than the severity of the proteostatic insult, at least for a while before levels of p90 were fully restored.

Levels of spliced *XBP1* mRNA likewise reached a maximum at ~6 hr of $\mu_s$ expression, and remained maximal for about 24 hr (with perhaps some oscillations) before they subsided to submaximal levels. Levels of CHOP, which provides a readout for PERK activity, reached a maximum slightly later, at 12–16 hr after the onset of $\mu_s$ expression, before reaching submaximal levels, as further illustrated in a 'heatmap' of UPR activation (*Figure 4D*). Treatment with ER stress-eliciting drugs during the chronic phase when the UPR had submaximal output readily triggered close to maximal output of all three UPR branches (*Figure 4—figure supplement 4*), indicating that prolonged $\mu_s$ expression had not exhausted the signaling capacity of the UPR, and thus, that homeostatic readjustment of the ER accounted for lowered UPR output levels upon chronic $\mu_s$ expression. Strikingly, the highest output of the three UPR branches did not coincide with $\mu_s$ having reached maximal expression levels. Instead, UPR signaling reached maxima when the ratio $\mu_s$/BiP was highest (*Figure 4B,D*). These

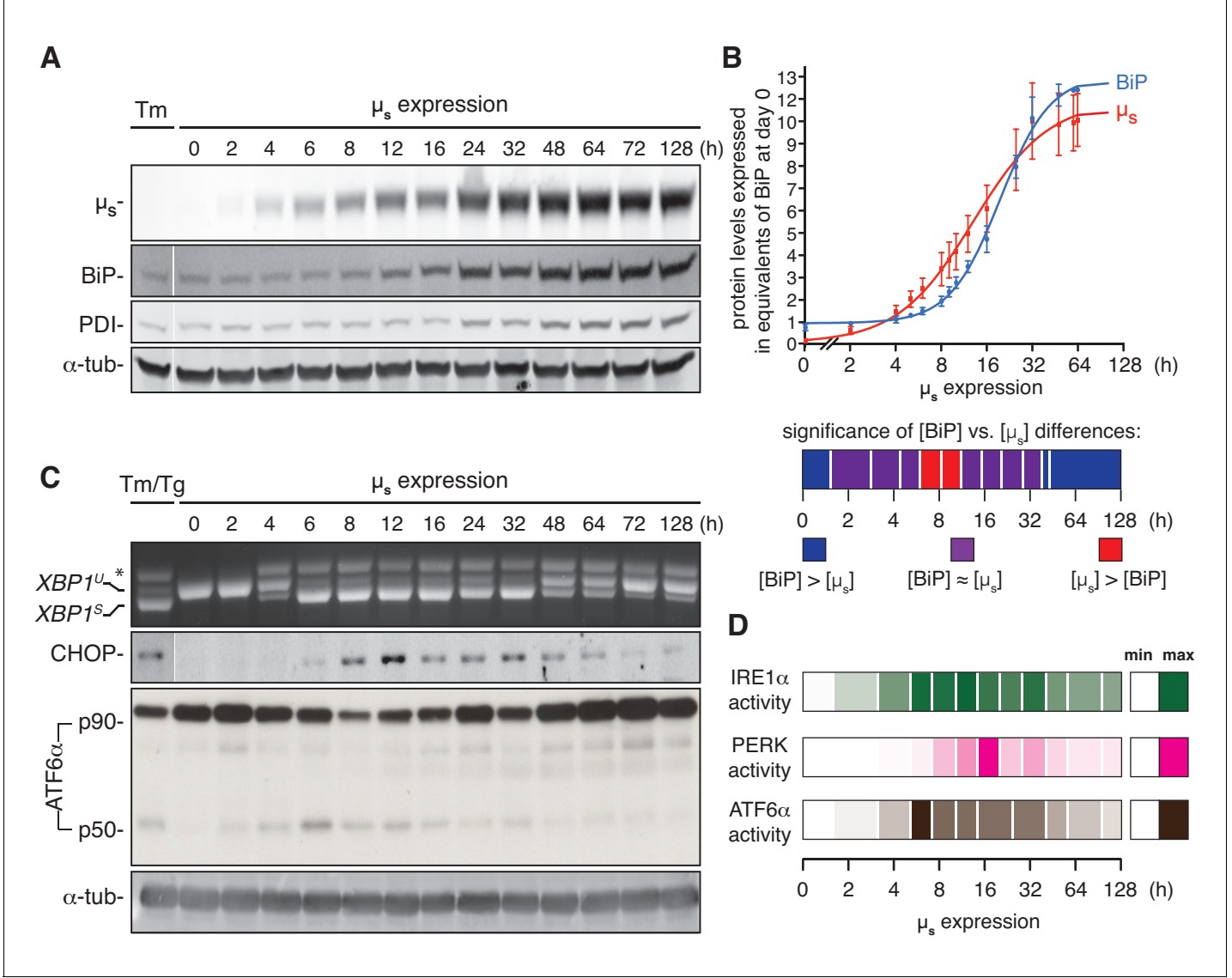

**Figure 4.** ER homeostatic readjustment entails a transition from acute to chronic UPR signaling. (A–C) HeLa-$\mu_s$ cells were induced with 0.5 nM Mif for various times as indicated. Levels of $\mu_s$, BiP, PDI (A) and $\alpha$-tubulin (A,C), as well as activation of the UPR pathways (C), were assessed as in **Figure 2A**. (B) Levels of $\mu_s$ and BiP were quantified using known quantities of BiP and $\mu_s$ as standards. The X-axis displays time in hrs in a logarithmic scale to show better detail of the early phase of the time course. Experiments were normalized to the level of BiP at $t = 64$ hr, which was reproducibly ~10 fold that at $t = 0$ hr. Quantities of BiP and $\mu_s$ are expressed in equivalents of BiP levels at $t = 0$ hr; mean and s.e.m. are shown, n = 5. Fitting was performed with Prism software to obtain a sigmoidal dose-response curve. As quantities were not assessed at the same time points in each experiment, some values were inferred from fit curves to obtain the s.e.m. Statistical significance of differences in expression levels between BiP and $\mu_s$ were tested by two-tailed t-test, and are depicted in a color-based heat map ranging from dark blue for [BiP] > [$\mu_s$] (***$p \leq 0.001$), via purple for when there is no statistical significance for differences between the levels of BiP and $\mu_s$, that is [BiP] $\approx$ [$\mu_s$], to red for [$\mu_s$] > [BiP] (*$p \leq 0.05$). (D) Activation of the UPR branches in (C) and replicate experiments were quantified (n = 2–20, depending on the time point assessed): for IRE1$\alpha$, the percentage of *XBP1* mRNA splicing; for PERK, the levels of expression of its downstream effector CHOP; and for ATF6$\alpha$, the levels of ATF6$\alpha$-p50 (both the latter normalized on the levels of $\alpha$-tubulin). Mean values of UPR signaling output at each time point were calculated as a percentage of the maximal level, and are depicted in a color-based heat map ranging from no color (min) to full color (max).

DOI: https://doi.org/10.7554/eLife.27518.010

The following source data and figure supplements are available for figure 4:

**Source data 1.** Data and calculations that were used to generate the bar graphs in *Figures 4B and D*.
DOI: https://doi.org/10.7554/eLife.27518.015
**Figure supplement 1.** Transitioning from acute, full-geared to chronic, submaximal IRE1$\alpha$ signaling upon $\mu_s$ expression.
DOI: https://doi.org/10.7554/eLife.27518.011

*Figure 4 continued on next page*

*Figure 4 continued*

**Figure supplement 2.** Transitioning from acute, full-geared to chronic, submaximal PERK signaling upon $\mu_s$ expression.
DOI: https://doi.org/10.7554/eLife.27518.012
**Figure supplement 3.** Transitioning from acute, full-geared to chronic, submaximal ATF6α signaling upon $\mu_s$ expression.
DOI: https://doi.org/10.7554/eLife.27518.013
**Figure supplement 4.** UPR pathways are not exhausted upon chronic $\mu_s$ overexpression.
DOI: https://doi.org/10.7554/eLife.27518.014

findings support a scenario in which the UPR signals are commensurate with the extent to which the folding machinery, in particular BiP, is sequested by $\mu_s$ rather than by $\mu_s$ accumulation per se.

## The ER expands in response to a proteostatic insult

The increase of BiP levels upon $\mu_s$ expression indicated that the homeostatic readjustment to accumulating $\mu_s$ levels entailed expansion of the ER. To directly visualize ER expansion, we targeted a modified version of pea peroxidase (APEX) (*Hung et al., 2014*; *Lam et al., 2015*) to the ER by use of an N-terminal signal peptide and a C-terminal KDEL. APEX-KDEL was expressed under control of the TetT promoter at such low levels that it did not interfere with the $\mu_s$-driven UPR (*Figure 5—figure supplement 1*). We then exploited APEX to catalyze polymerization of 3,3'-diaminobenzidine tetrahydrochloride (DAB) upon treatment with $H_2O_2$ to obtain contrast in electron microscopy (EM), and stained the ER of cells before or after 1, 3, or 7 days of $\mu_s$ expression.

We determined that, before induction of $\mu_s$ expression, the ER lumen occupies 10–12% of the area within the cytoplasm (i.e. excluding the nucleus) in the electron micrographs (*Figure 5*), which by a rough estimate would correspond to $(0.10–0.12)^{3/2} \approx 3–4\%$ of the cytoplasmic volume. Considering that the volume of the cytosol in HeLa cells is estimated to be $\sim 2 \times 10^3$ $\mu m^3$, that is $\sim 2 \times 10^3$ fl (*Milo, 2013*), we estimated the ER to have a volume of ~60–80 fl. The ER expanded in the course of 1–3 days upon the onset of $\mu_s$ expression to cover ~18% of the cytoplasmic area in the electron micrographs, corresponding to roughly $(0.18)^{3/2} \approx 7–8\%$ of the cytoplasmic volume, but we observed no further ER expansion after 3 days. On the basis of these estimates, the volume of the ER increased 2–3-fold as a result of $\mu_s$ expression, to ~120–240 fl per cell.

## The ER becomes dominated by the chaperone BiP in response to a proteostatic insult

We further corroborated the $\mu_s$-driven ER expansion by proteomic analysis and identified ~3000 proteins from lysates of cells before or after 1, 3, or 7 days of $\mu_s$ expression. We then used a label-free quantitation approach to obtain an approximation of the mass levels of proteins (*Cox et al., 2014*) (*Supplementary file 1*), such that we could estimate what share of the total protein mass content of the cell the ER would account for. Proteins that we identified as ranking among the 500 most abundant at any of the time points, altogether representing ~600 proteins, were categorized according to subcellular localization and/or function. We limited detailed analysis to this subset, because the label-free quantitation method is more reliable for abundant proteins (*Cox et al., 2014*), and because this subset already covered >90% of the total protein content in the cell. Changes in protein expression levels were assessed based on the label-free quantitations and in parallel by SILAC (*Ong et al., 2002*), which revealed mostly similar trends in differential expression (*Supplementary file 1*) and confirmed the usefulness of the label-free quantitation approach.

We then calculated the percentage of the total protein for each category and deduced that the ER accounts for ~3% of total protein content before $\mu_s$ expression in line with the findings by EM. Upon $\mu_s$ expression, the ER content, including $\mu_s$, expanded to ~10% within 3 days, but did not expand much further after that (~12% at day 7) (*Figure 6*, upper panels; *Supplementary file 1*). The estimate of the percentage of the total protein that the ER accounts for that was obtained by proteomics was slightly higher than the EM-based estimate of the percentage of the total volume that the ER occupies within the cell, possibly indicating that continuous $\mu_s$ expression led to increased molecular crowding within the ER. In contrast to the ER, other organelles or cellular machineries were not markedly affected by $\mu_s$ expression (*Supplementary file 1*).

Levels of ER resident chaperones underwent drastic changes upon $\mu_s$ expression (*Figure 6*, lower panels; *Supplementary file 1*). Most notably, between day 0 and 7, levels of BiP increased ~8-fold

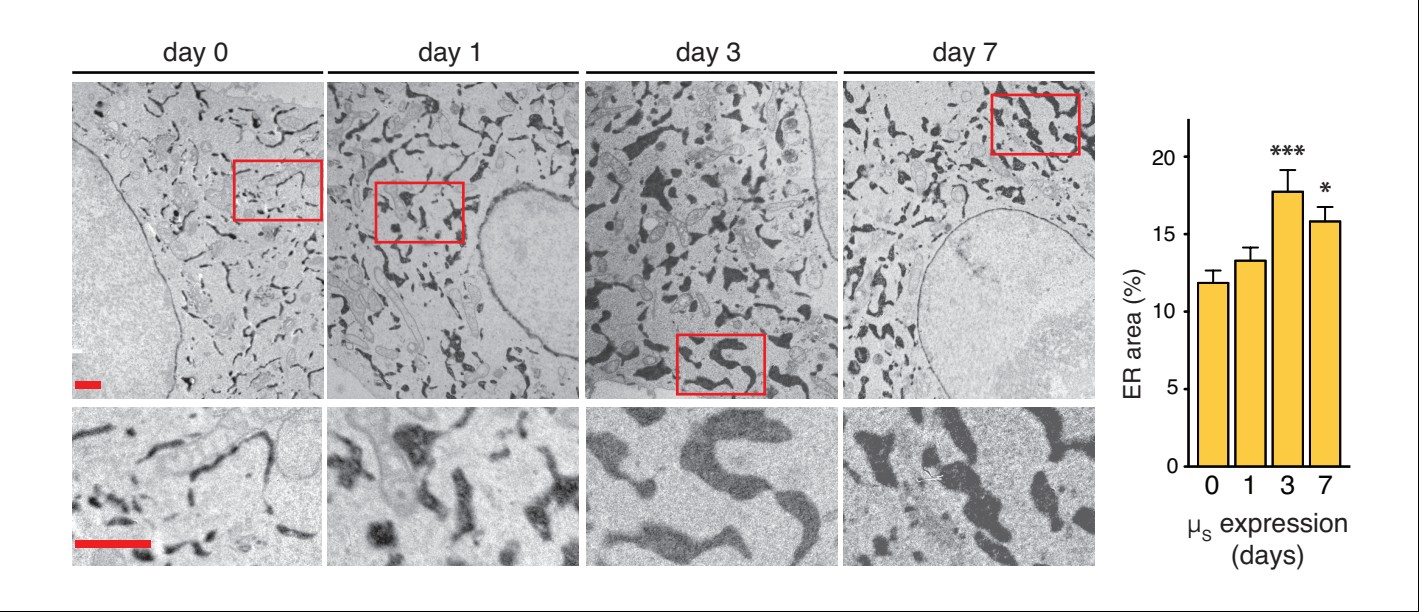

**Figure 5.** ER expansion in the course of homeostatic readjustment to $\mu_s$ overexpression. HeLa-$\mu_s$-derived cells, harboring Dox-inducible APEX-KDEL, were induced with 0.5 nM Mif to express $\mu_s$ for various days as indicated, and APEX-KDEL expression was induced with 100 nM Dox for 2 days. APEX-KDEL was exploited to obtain DAB precipitates (dark), revealing the extent of the ER in electron micrographs. Boxed areas are shown by 3-fold magnification; scale bars represent 1 μm. The percentage of the area within the cytoplasm corresponding to ER was determined and depicted in bar graphs; mean and s.e.m. are shown, n = 10. Statistical significance of differences in the extent of ER occupying cytosolic area in the electron micrographs was tested by ANOVA (*$p \leq 0.05$; ***$p \leq 0.001$).

DOI: https://doi.org/10.7554/eLife.27518.016

The following source data and figure supplement are available for figure 5:

**Source data 1.** Data and calculations that were used to generate the bar graph.
DOI: https://doi.org/10.7554/eLife.27518.018

**Figure supplement 1.** APEX-KDEL expression hardly interferes with $\mu_s$-driven UPR.
DOI: https://doi.org/10.7554/eLife.27518.017

(and ~14-fold according to SILAC measurements; *Supplementary file 1*), in good agreement with the absolute quantitations we obtained by immunoblotting (*Figure 4B*). In fact, at day 7, BiP became the second most abundant protein in the cells (after β-actin), amounting to ~4% of the total proteome mass. Put together with the absolute quantitation of BiP levels, we estimate that the cells harbor an average of $5 \times 10^9$ proteins, which fits with earlier assumptions (*Milo, 2013*). Moreover, from these quantitations, we estimated BiP levels to be at ~0.5 mM (or 30–40 mg/ml) before induction and to reach ~1.5 mM (or 90–120 mg/ml) in the ~3-fold expanded ER at day 7, accounting for about a third of the estimated ~200–300 mg/ml overall protein concentration in the lumen of the organelle. Other ER resident chaperones, such as GRP94, CRT and some members of the PDI family (most notably ERp72 and P5), also increased albeit to a lesser extent. As a consequence, levels of BiP within the ER increased from ~15% before the onset of $\mu_s$ expression to 30–40% of the expanded ER proteome. Levels of GRP170, which acts as a nucleotide exchange factor of BiP (*Behnke et al., 2015*), also increased markedly. The ERdj co-chaperones of BiP were upregulated upon $\mu_s$ expression as well, but, surprisingly perhaps, to levels that were still about two orders of magnitude lower than the levels of BiP itself (*Supplementary file 1*). Thus, their co-chaperone function must be executed in a sub-stoichiometric fashion in the HeLa-$\mu_s$ model.

Before induction, $\mu_s$ was barely detectable (~1 ppm) but levels rapidly increased to become ~$10 \times 10^3$ ppm, that is ~1% (after 1 day) and ~2% (from day 3 onward) of the total protein content in the cell (*Figure 6*, upper panels; *Supplementary file 1*). Strikingly, 1 day of expression was sufficient for $\mu_s$ levels to exceed that of any chaperone in the ER except those of BiP, which still outmatched $\mu_s$ 3:2 (*Figure 6*, lower panels; *Supplementary file 1*). Yet, BiP levels had increased 3–

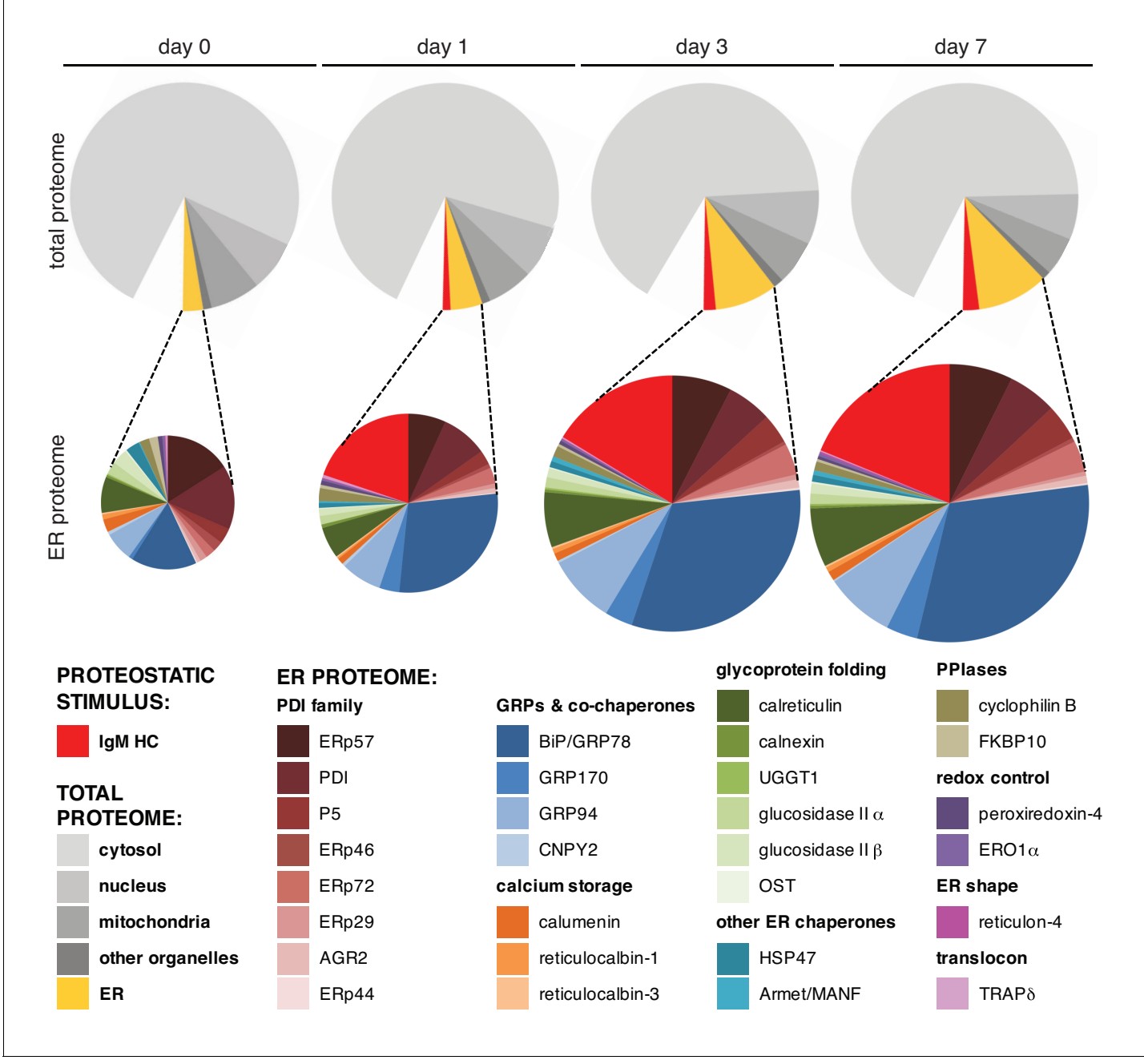

**Figure 6.** BiP becomes dominant in the ER when it expands upon μ$_s$ overexpression. HeLa-μ$_s$ cells were induced with Mif to express μ$_s$ for various numbers of days as indicated. Proteins were identified at each time point by mass spectrometry and expression levels were approximated by label-free proteomics. The ~600 most abundant proteins, accounting for over 90% of the total protein content, were categorized according to localization in the cell—cytosolic, nuclear, mitochondrial, ER resident, or residing in other organelles—by the use of Uniprot entries. From the approximated quantities of these proteins, the quantities of proteins per organelle were calculated as a percentage of the total proteome and depicted in pie charts (upper panels). For the ER, the resident proteins (gold) and μ$_s$ (red) are shown separately. Quantities of ER resident proteins including μ$_s$ were calculated as a percentage of the ER proteome and are depicted in pie charts (lower panels) that are size-proportioned commensurate with the difference in combined quantity of ER and μ$_s$ at different days. For details on calculations, see *Supplementary file 1*. Color-coding of pie charts is as annotated in the legend embedded in the figure.

DOI: https://doi.org/10.7554/eLife.27518.019

The following source data is available for figure 6:

**Source data 1.** Data derived from *Supplementary file 1* that were used to generate the pie diagrams.
DOI: https://doi.org/10.7554/eLife.27518.020

4-fold during that first day to keep pace with the burden imposed on the ER folding machinery by $\mu_s$. At later times, the margin by which BiP levels were in excess of $\mu_s$ levels increased again, with BiP and $\mu_s$ together constituting about half of the ER protein mass when it had fully expanded. Thus, upon $\mu_s$ expression in bulk, the ER content not only changed quantitatively (i.e. undergoing a ~ 3–4-fold expansion), but also qualitatively (i.e. becoming dominated by $\mu_s$ that is held in check by excess levels of its most devoted chaperone BiP).

### UPR signaling is key for ER homeostatic readjustment

As cell growth and survival were hardly perturbed upon $\mu_s$ expression, we reasoned that ER homeostasis was successfully readjusted. The HeLa-$\mu_s$ model therefore permitted to ask the key question of whether UPR activation is essential for ER homeostatic readjustment. As we have shown above for wild-type HeLa cells (*Figure 1D*) ablation of all three UPR branches together hardly impeded cell growth in HeLa-$\mu_s$ cells in the absence of $\mu_s$ expression (*Figure 7A,B*), probably because HeLa cells by default handle a low secretory load (*Supplementary file 1*). Concomitant expression of $\mu_s$, however, fully abolished cell growth (*Figure 7A,B*). Annexin V staining confirmed that $\mu_s$ expression caused synthetic lethality under conditions when all UPR branches were ablated (*Figure 7C*). Thus, we concluded that the UPR is essential for ER homeostatic readjustment upon $\mu_s$ expression.

### Discussion

By avoiding the use of ER stress-eliciting drugs that have cytotoxic side effects, we have created a cell model that allowed us to reevaluate several aspects of how cells respond to an accumulating load of client protein in the ER. First, our results highlight that the UPR predominantly acts as a pro-survival pathway. By confronting cells with a proteostatic stimulus (i.e. bulk $\mu_s$ expression) that these cells could well cope with, we confirmed that the UPR is key for their capacity to do so. Yet, the finding that $\mu_s$ expression readily causes cells to undergo apoptosis in the absence of these UPR transducers implies that other, as yet unknown, ER stress-induced mechanisms can be invoked to mediate cell death. These mechanisms may well play such a role also when the UPR is functional. For instance, hyperoxidizing conditions in the lumen of the ER activate NADPH oxidases, leading to

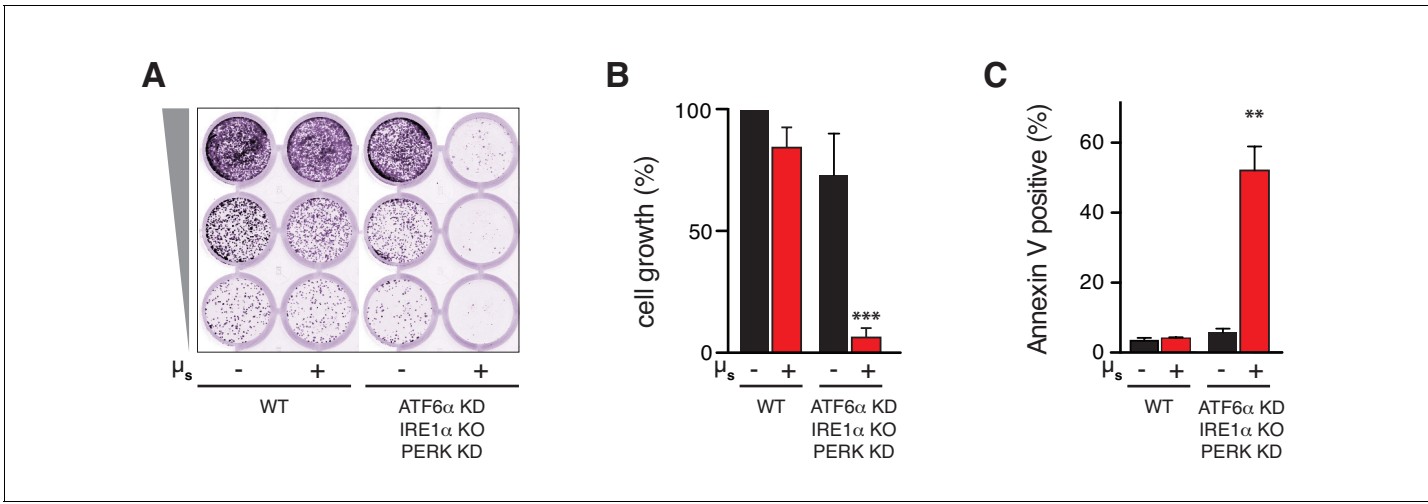

**Figure 7.** The UPR is essential to sustain ER homeostatic readjustment upon $\mu_s$ expression. (A,B,C) All UPR transducers were ablated, either by deletion (KO, IRE1$\alpha$) or by silencing (KD, ATF6$\alpha$ and PERK) in HeLa-$\mu_s$ cells, and $\mu_s$ expression was induced with 0.5 nM Mif (+) or not (–), as indicated. (A,B) Cell growth was assessed as in *Figure 1B* and quantitated as in *Figure 1C* (B). Mean and s.e.m. are shown in a bar graph; n = 2. (C) Percentages of Annexin-V-positive cells before or upon induction of $\mu_s$ expression for 3 days were assessed by cytometric analysis. Means and s.e.m. are shown in a bar graph, n = 2. (B,C) Statistical significance of differences in growth (B) or Annexin V staining (C) were tested by ANOVA (**$p \leq 0.01$; ***$p \leq 0.001$).
DOI: https://doi.org/10.7554/eLife.27518.021

The following source data is available for figure 7:

**Source data 1.** Data and calculations that were used to generate the bar graphs in *Figures 7B and C*.
DOI: https://doi.org/10.7554/eLife.27518.022

efflux of radical oxygen species to the cytosol that may cause activation of pro-apoptotic pathways (*Tabas and Ron, 2011*).

Second, our data confirm that the accumulation of client proteins in the ER drives ER expansion. Interestingly, ER expansion is not necessarily provoked by bulk synthesis of client proteins per se, as co-expression of sufficient quantities of λ led to bulk secretion of IgM but failed to trigger a strong UPR, and, thus, did not result in augmented BiP production (which served as a proxy for ER expansion). Our findings indicate moreover that the ER folding machinery seems to be underused in HeLa cells. Apparently, mobilization of the default ER folding machinery is sufficient for a task that can be productively executed, such as the folding and assembly of IgM from subunits that are expressed in bulk but in the correct stoichiometry. In most cells, the basal size of the ER and its folding capacity may indeed exceed the need for folding assistance from the default secretory burden. In line with such a notion, a substantial fraction of BiP is kept in reserve in an inactive, AMPylated form under non-ER stress conditions (*Preissler et al., 2015*).

Third, our results highlight the key role of BiP in ER homeostasis. In fact, our choice to employ $\mu_s$ as a proteostatic insult to challenge the ER was motivated in part by the reasoning that if there were ever an ER client protein that would be excellent in sequestering BiP, it would be $\mu_s$ (*Haas and Wabl, 1983*; *Bole et al., 1986*). Furthermore, it is well-conceivable that even the slightest leakiness of unpaired $\mu_s$ from the ER has been evolutionarily selected against. As antigen recognition is likely to be compromised if $\mu_s$ is released from the cell when unaccompanied by λ, such a release would potentially lead to ill-fated off-target effects. Thus, BiP binding to $\mu_s$ must be highly stringent out of immunological necessity (*Anelli and van Anken, 2013*), which correlates with the fact that mutations in $\mu_s$ that lead to lowered affinity for BiP are associated with disorders of the immune system, such as lymphoproliferative heavy chain disease or myeloma (*Hendershot et al., 1987*; *Anelli and Sitia, 2010*).

To ascertain stringent ER retention of the accumulating load of $\mu_s$, the pool of BiP and the ER at large expanded to an impressive extent, echoing ER expansion during B cell to plasma cell development. In the course of B to plasma cell differentiation, however, all cellular machineries develop in a strict sequential order that allows them to anticipate bulk IgM production (*van Anken et al., 2003*). Conversely, the purely $\mu_s$-driven stimulus that we investigated in this study resulted almost exclusively in ER expansion. We found only meager indications of the expansion of even the remainder of the secretory machinery (*Supplementary file 1*). One could argue that the stringent retention of $\mu_s$ in the ER by BiP would obviate the need to reinforce the secretory pathway downstream of the ER. Accordingly, homeostatic regulation of the Golgi and compartments beyond appears to be governed foremost by dedicated signaling mechanisms distinct from the UPR (*Luini and Parashuraman, 2016*).

Importantly, we found that maximal UPR activation correlated with a shortage of BiP in matching the accumulating load of $\mu_s$, rather than with accumulation of client protein in the ER per se. Indeed, the transitioning of UPR signaling to chronic submaximal levels coincided with the ER becoming dominated by BiP. Our findings are in line with the idea—which was first coined by David Ron and colleagues (*Bertolotti et al., 2000*)—that sequestering of BiP by ER clients (such as $\mu_s$) is key for activating the UPR, as activation of the main three UPR sensors (IRE1α, PERK [*Bertolotti et al., 2000*] and ATF6α [*Shen et al., 2002*]) goes hand in hand with BiP dissociation from their respective luminal ER stress-sensing domains, whereas overexpression of BiP dampens UPR activation (*Bertolotti et al., 2000*). Accordingly, despite its accumulation in the ER, the PiZ mutant of $\alpha_1$-antitrypsin does not sequester BiP and fails to trigger the UPR (*Graham et al., 1990*). An in vitro study moreover suggests that sequestering of BiP by $\mu_s$ would be sufficient to activate IRE1α and PERK (*Carrara et al., 2015*). However, based on the crystal structure of yeast Ire1's lumenal domain (*Credle et al., 2005*), Peter Walter and colleagues proposed instead that Ire1 activates through direct binding of unfolded proteins, for which in vitro evidence has subsequently been obtained for both yeast Ire1 (*Gardner and Walter, 2011*) and mammalian IRE1α (*Karagöz et al., 2017*).

Assuming that the UPR signaling amplitude is commensurate with that of ER stress sensing by the UPR transducers, it is tempting to conclude that the UPR transducers are in the OFF state (as ER stress sensors) when bound to BiP, and in the ON state when bound to clients. So in a three-way competition between formation of sensor-BiP, sensor-client, and BiP-client complexes, the ratio between sensor-client (ON) and sensor-BiP (OFF) will most robustly report on the ratio between

client and BiP. Thus, the two models for ER stress sensor activation can be reconciled easily in a model of ratiometric ER stress sensing, in line with our observations that the UPR signaling output is highest when the levels of the ER client $\mu_s$ eclipse those of BiP, and that UPR signaling subsides to submaximal levels when an excess of BiP over the client is restored.

The finding that homeostatic readjustment of the ER entails that the UPR output becomes submaximal is also relevant for disease. Various genetic disorders stem from mutations in ER client proteins that lead to their misfolding and accumulation in the ER. On the basis of our results, cells that suffer from such conditions will likely adapt and display submaximal UPR signaling levels similar to those that we found for chronic $\mu_s$ expression. In other words, submaximal UPR signaling is not a sign of 'mild' ER stress, but rather reflects that the stress is chronic.

Finally, since successful ER homeostatic readjustment upon $\mu_s$-driven proteostatic ER stress critically depends on the UPR, the HeLa-$\mu_s$ cell model provides an excellent tool to dissect in greater detail how the UPR alleviates ER stress, and how it may serve to evaluate the success of ER homeostatic readjustment. The latter aspect is key, because a variety of disorders, ranging from cancer to diabetes, tie in with aberrant or maladaptive UPR-driven cell fate decisions (*Wang and Kaufman, 2012*).

## Materials and methods

### Cell culture and Reagents

All reagents were obtained from Sigma-Aldrich, Milan, Italy, unless otherwise stated. HeLa S3 cells, of which the genotype was confirmed by PCR single locus technology, and all derivate lines (*Supplementary file 2A*) were cultured in DMEM (Thermo Fischer Scientific, Monza, Italy) containing glutamax (1 mM), 5% Tet-System approved Fetal Bovine Serum (FBS, Takara, Jesi, Italy), 100 U/ml penicillin and 100 µg/ml streptomycin. Cells were routinely tested (on a monthly basis) and found to be mycoplasm-free by use of a standard diagnostic PCR. Expression of transgenes was induced with 0.5 nM Mif unless indicated otherwise, and/or Dox at various concentrations as indicated. I.29µ⁺ lymphomas were used as model B lymphocytes (*Alberini et al., 1987*), and cultured in suspension in RPMI (Thermo Fischer Scientific) supplemented with 10% LPS free FBS (GE Healthcare, Milan, Italy), glutamax (1 mM), penicillin (100 U/ml), streptomycin (100 µg/ml), sodium pyruvate (1 mM) and β-mercaptoethanol (50 µM). I.29µ⁺ cells were induced to differentiate with 20 µg/ml LPS.

### Generation of HeLa-derived cell lines that inducibly express transgenes

The pSwitch cassette (GeneSwitch system; Thermo Fischer Scientific), placed into a lentiviral vector as described (*Sirin and Park, 2003*)—a kind gift from Dr Frank Parker—was used to render cells Mif-responsive, as the pSwitch cassette encodes a hybrid nuclear receptor that is activated by Mif to drive expression of genes under the control of the GAL4 promoter. Similarly, a reverse tetracycline-dependent transactivator (rtTA) cassette (*Zhou et al., 2006*), under control of a bidirectional promoter with ΔLNGFR in the reverse orientation for selection purposes (*Amendola et al., 2005*), in a lentiviral vector was employed to render cells Tet (and thus Dox)-responsive. A pGene5b (GeneSwitch system) encoding lentiviral construct (*Sirin and Park, 2003*)—another kind gift from Dr Frank Parker—was used as backbone to place either Igµ$_s$ or Igλ under control of the GAL4 promoter. The coding sequences for NP-specific murine $\mu_s$ and λ were derived from plasmids described elsewhere (*Sitia et al., 1987*; *Fagioli and Sitia, 2001*). pENTR223.1 #100061599 bearing murine IRE1α cDNA was obtained from Thermo Fisher Scientific. GFP was placed in frame at the same position in the juxtamembrane cytosolic linker portion of IRE1α as described (*Li et al., 2010*). The IRE1α-GFP cassette was placed under control of a TetTight (TetT)-responsive element (Takara) in a lentiviral vector. The APEX2 coding sequence (*Lam et al., 2015*), derived from the #49385 plasmid (Addgene, Cambridge, MA, USA), and modified to contain in frame extensions encoding the vitronectin signal peptide at the N-terminus and the tetrapeptide KDEL at the C-terminus, was placed under control of a TetT promoter in the same lentiviral vector as employed for IRE1α-GFP. Standard techniques were used for construction, transformation and purification of plasmids. Transgene cassettes were genomically integrated in a subsequent manner into HeLa S3 cells by lentiviral delivery, essentially as described (*Amendola et al., 2005*). Cells with genomic integrations of transgenes were cloned by limiting dilution to yield the cell lines used in this study as summarized in (*Supplementary file 2A*).

## Ablation of UPR pathways, and pharmacological activation of the UPR

CRISPR/Cas9 was used to delete endogenous IRE1$\alpha$ in HeLa S3 or derivate cells (*Supplementary file 2A*), and clones were obtained by limiting dilution and verified by their lack of *XBP1* mRNA splicing upon Tm treatment. Gene silencing of ATF6$\alpha$, IRE1$\alpha$, PERK, and XBP1 was performed with pooled ON-TARGET plus siRNA (GE Healthcare) (*Supplementary file 2C*) according to the manufacturers' instructions. Cells were seeded the day after transfection with the siRNA pools. The UPR was pharmacologically elicited with either 300 nM Tg or 5 µg/ml (or lower) Tm.

### *XBP1* mRNA splicing assay

Total RNA was extracted from cells using the UPzol RNA lysis reagent (Biotechrabbit, Hennigsdorf, Germany) followed by a standard protocol provided by the manufacturer for assessing *XBP1* mRNA splicing levels: cDNA was obtained from samples and amplified by PCR. Oligos used to amplify cDNA corresponding to *XBP1* mRNA have been described (*Calfon et al., 2002*). PCR products were resolved on agarose gels and images were acquired with a Typhoon FLA-9000 reader (GE Healthcare). The percentage of spliced *XBP1* mRNA was calculated as described (*Shang and Lehrman, 2004*).

## Cell growth assay

To assess growth, cells were counted with a Burker chamber, and seeded in 24-multiwell plates in 1:5 serial dilutions (5000, 1000, and 200 cells per well) to grow for 7 days. Culture media and pharmacological agents were added as soon as cells attached after plating and refreshed every 2–3 days. Cells were fixed in methanol-acetone (1:1) for 10 min, stained with 0.5% crystal violet in 20% methanol for 10 min and washed with distilled water. Dried plates were densitometrically scanned at a resolution of 50–100 µm with the Typhoon FLA-9000 reader, employing the 647 nm laser and the photomultiplier 1000. Intensity of crystal violet staining was analyzed with ImageJ for quantitation of growth. Typically, wells seeded with 1000 cells were used for comparison of signal intensity between conditions. An empty well on the same plate served for background subtraction.

## Protein analysis

Protein extraction, sample preparation, electrophoresis on 10% or 4–12% Bis/Tris precast polyacrylamide gels (Thermo Fischer Scientific), and transfer of proteins onto nitrocellulose (GE Healthcare) for immunoblot analysis with antibodies listed in *Supplementary file 2B* were performed using standard techniques with the following exceptions: for analysis of CHOP and ATF6$\alpha$ cells, total lysates were used to ensure that the nuclear pools of these proteins were solubilized; for analysis of disulfide-linked IgM assembly intermediates, reducing agents were omitted from the lysis buffer and free sulfhydryl groups were alkylated with N-ethyl-maleimide (NEM); for analysis of ATF6$\alpha$, 1.5 mm thick 8% polyacrylamide gels, prepared from a 30% polyacrylamide-bis 29:1 solution (Biorad, Milan, Italy), were used as described previously (*Maiuolo et al., 2011*). Protein transfer onto nitrocellulose was confirmed by reversible Ponceau staining. Protein quantitation of samples was performed using a bicinchoninic acid.

   Detection of fluorescent antibody signals on blots was performed by scanning with the Typhoon FLA-9000 reader, except for IRE1$\alpha$ and ATF6$\alpha$, which were detected on films using HRP-conjugated secondary antibodies and ECL. Signal intensities were analyzed with Image J. For absolute quantitation of BiP and $\mu_s$ protein levels, standard $\mu_s$ and BiP curves were obtained. To that end, we used purified mouse-myeloma-derived IgM (#02–6800, Thermo Fischer Scientific), of which we estimated 70% of the protein mass to be $\mu_s$, and purified recombinant hamster BiP, which save for two conservative changes, Y313F and A649S, is identical in sequence to human BiP and therefore, in all likelihood, is recognized equally well by the goat anti-BiP antibody (C-20; Santa Cruz, Heidelberg, Germany). Recombinant BiP was expressed and purified as described (*Marcinowski et al., 2011*) from a pPROEX expression construct that was a kind gift from Dr Johannes Buchner. To avoid potential crossreaction of secondary antibodies against $\mu_s$, we used an anti-IgM antibody that was itself Alexa-546-conjugated to allow fluorescent detection.

## Radiolabeling

Cells were starved for 10 min in standard medium, containing 1% dialyzed FBS but lacking cysteine and methionine, prior to 10 min pulse labeling with Express [$^{35}$S] Protein labeling mix (Perkin Elmer, Milan, Italy) containing 40 µCi $^{35}$S-methionine and $^{35}$S-cysteine per $10^6$ cells. Cells were harvested and washed twice in ice-cold HBSS (Thermo Fischer Scientific). After a wash in ice-cold PBS, cells were lysed in RIPA buffer containing NEM and protease inhibitors for 10 min on ice, as described previously (*Fagioli and Sitia, 2001*). Lysates were cleared for 10 min at 13,000 rpm at 4°C. Before immunoprecipitation, cell lysates were pre-cleared for 1 hr with 30 µl FCS-Sepharose (GE Healthcare) and µ$_s$ was immunoprecipitated for 16 hr using a rabbit anti-mouse IgM (H) antibody (#61–6800, Life Technologies). Immunoprecipitates were collected on protein G-Agarose beads (Thermo Fischer Scientific), washed twice in 10 mM Tris (pH 7.4), 150 mM NaCl and 0.5% NP-40, and once in 5 mM Tris-HCl (pH 7.5), before gel electrophoresis. Gels were transferred and dried onto a 3 MM filter paper, and exposed to a LE storage phosphor screen (GE Healthcare). Signals were acquired on the Typhoon FLA-9000 with a phosphorimaging filter. Densitometric analysis of signals was performed with ImageJ.

## Fluorescence microscopy

Samples were prepared for immunofluorescence as described previously (*Sannino et al., 2014*), and sample-containing coverslips were mounted on glass slides with Mowiol. Light microscopic images were acquired with an UltraView spinning disc confocal microscope operated by Volocity software (PerkinElmer). Images were processed with Photoshop (Adobe, San Jose, CA, USA). Antibodies used were: Alexa-546 anti-mouse IgM (µ) 1:1000 (Life Technologies, Monza, Italy); rabbit anti-calreticulin 1:200, and secondary Alexa-488-anti-rabbit 1:300 (Life Technologies). Nuclei were stained with DAPI.

## Apoptosis assay

As a measure for apoptosis, the percentage of Annexin-V-positive cells was recorded by use of a standard detection assay (APC-Annexin V, BD Biosciences, Milan, Italy) on a Canto cytometer (BD Biosciences) following the manufacturers' instructions.

## Electron microscopic determination of ER size

Cells harboring APEX2-KDEL were fixed in 1% glutaraldehyde, 0.1 M sodium cacodylate (pH 7.4) for 30 min, and incubated for 20 min with 0.3 mg/ml DAB, and for 20 min with 0.03% H$_2$O$_2$ (to activate APEX) in 0.1 M sodium cacodylate (pH 7.4), all at room temperature. Samples were rinsed in 0.1 M sodium cacodylate (pH 7.4), and post-fixed with 1.5% potassium ferrocyanide, 1% OsO$_4$, sodium cacodylate (pH 7.4) for 1 hr on ice. After en bloc staining with 0.5% uranyl acetate overnight at 4°C in the dark, samples were dehydrated with increasing concentrations of ethanol, embedded in EPON and cured in an oven at 60°C for 48 hr. Ultrathin sections (70–90 nm) were obtained using an ultramicrotome (UC7, Leica microsystem, Vienna, Austria), collected, stained once more with uranyl acetate and Sato's lead solutions, and visualized in a transmission electron microscope (Leo 912AB, Carl Zeiss, Oberkochen, Germany). Digital micrographs were taken with a 2K × 2K bottom-mounted slow-scan camera (ProScan, Lagerlechfeld, Germany) controlled by EsivisionPro 3.2 software (Soft Imaging System, Münster, Germany). Images of randomly selected cells (10 for each condition) were acquired at a nominal magnification of 1840x. Using ImageJ software, cytoplasmic regions were selected manually and ER profiles were segmented, thanks to the electron-dense DAB precipitate, by means of an automatic macro based on threshold intensity, yielding the percentage of the area within the cytoplasm that was ER. For simplicity, we used the formula: volume (%) = (area (%))$^{3/2}$—which would be fully accurate only if the ER were cuboidal, but which we considered a good approximation nonetheless—to obtain a rough estimate of the ER volume.

## Proteomics analysis

SILAC heavy and light collected serum-free cell media were mixed proportionally to a 1:1 cell number ratio and lysed with 8 M Urea 10 mM Tris-HCl buffer. Proteins were then quantified by Bradford. About 5 µg of mixed proteins for each sample were reduced by TCEP, alkylated by

chloroacetamide, and digested by Lys-C and trypsin (*Kulak et al., 2014*), before peptides were desalted on StageTip C18 (*Rappsilber et al., 2003*).

Samples were analyzed in duplo on a LC–ESI–MS-MS quadrupole Orbitrap QExactive-HF mass spectrometer (Thermo Fisher Scientific). Peptides separation was achieved on a linear gradient from 93% solvent A (2% ACN, 0.1% formic acid) to 60% solvent B (80% acetonitrile, 0.1% formic acid) over 110 min,and from 60% to 100% solvent B in 8 min at a constant flow rate of 0.25 μl/min on UHPLC Easy-nLC 1000 (Thermo Fischer Scientific) connected to a 23 cm fused-silica emitter of 75 μm inner diameter (New Objective, Inc. Woburn, MA, USA), packed in-house with ReproSil-Pur C18-AQ 1.9 μm beads (Dr Maisch Gmbh, Ammerbuch, Germany) using a high-pressure bomb loader (Proxeon, Odense, Denmark). MS data were acquired using a data-dependent top 20 method for HCD fragmentation. Survey full scan MS spectra (300–1650 Th) were acquired in the Orbitrap with 60,000 resolution, AGC target $3^{e6}$, IT 20 ms. For HCD spectra, resolution was set to 15,000 at $m/z$ 200, AGC target $1^{e5}$, IT 80 ms; NCE 28% and isolation width 2.0 $m/z$.

For identification and quantitation, raw data were processed with MaxQuant version 1.5.2.8 searching against the database uniprot_cp_human_2015_03 + sequences of $\mu_s$ and the Mif respon-sive hybrid nuclear receptor Switch setting labeling Arg10 and Lys8, trypsin specificity and up to two missed cleavages. Cysteine carbamidomethyl was used as fixed modification, methionine oxidation and protein N-terminal acetylation as variable modifications. Mass deviation for MS-MS peaks was set at 20 ppm. The peptides and protein false discovery rates (FDR) were set to 0.01; the minimal length required for a peptide was six amino acids; a minimum of two peptides and at least one unique peptide were required for high-confidence protein identification.

The lists of identified proteins were filtered to eliminate known contaminants and reverse hits. Normalized H/L ratios were analyzed via Perseus (version 1.5.0.6). Statistical analysis was performed using the Significance B outlier test where statistical significance based on magnitude fold-change was established at p<0.05. To look for proteins that changed over time, we considered Intensity L and Intensity H normalized by the correspondent Summed Intensity and the statistical ANOVA test analysis was performed using Perseus (ver. 1.5.0.6) with p<0.01. All proteomic data as raw files, total proteins, and peptides identified with relative intensities and search parameters have been loaded into Peptide Atlas repository (ftp://PASS01009:PJ3566i@ftp.peptideatlas.org/). To obtain approxi-mations for absolute protein quantities, we followed a MaxLFQ label-free quantification strategy, as described previously (*Cox et al., 2014*). We then assessed the relative abundance of total protein per subcellular compartment, that is cytosol, nucleus, mitochondrion, ER, and other organelles, as well as $\mu_s$, as well as the relative abundance of protein species, including or excluding $\mu_s$, within the ER as detailed in the legend of *Supplementary file 1*.

## Acknowledgements

We thank Drs Bernhard Gendtner, Frank Park, and Johannes Buchner for sharing plasmids. We thank Dr Caterina Valetti for her advice on immunoblotting of ATF6α, Dr John Christiansson for help with RNA silencing, Dr Andrew Bassett for gRNA design, and Dr Stefano Bestetti for assistance with FACS analysis. All members of the ALEMBIC imaging facility and the Anna Rubartelli, Paola Panina, Sitia, Bachi and Van Anken labs are acknowledged for stimulating discussions and advice. We thank Drs Maurizio Molinari, Alison Forrester, John Christiansson, Tomás Aragón, and Luca Rampoldi for proofreading of the manuscript.

## Additional information

### Funding

| Funder | Grant reference number | Author |
| --- | --- | --- |
| Giovanni Armenise-Harvard Foundation | | Eelco van Anken |
| Ministero della Salute | RF-2011-02352852 | Eelco van Anken |
| Associazione Italiana per la Ricerca sul Cancro | MFAG 13584 | Eelco van Anken |

| Ministero della Salute | PE-2011-02352286 | Roberto Sitia<br>Eelco van Anken |
| Associazione Italiana per la Ricerca sul Cancro | IG 2016-18824 | Roberto Sitia |
| Fondazione Telethon | GGP15059 | Roberto Sitia |
| Fondazione Cariplo | 2015-0591 | Roberto Sitia |

The funders had no role in study design, data collection and interpretation, or the decision to submit the work for publication.

### Author contributions

Anush Bakunts, Andrea Orsi, Milena Vitale, Conceptualization, Data curation, Formal analysis, Supervision, Validation, Investigation, Visualization, Methodology, Writing—review and editing; Angela Cattaneo, Data curation, Investigation; Federica Lari, Laura Tadè, Investigation; Roberto Sitia, Methodology, Funding acquisition; Andrea Raimondi, Investigation, Visualization, Methodology, Writing—review and editing; Angela Bachi, Conceptualization, Data curation, Supervision, Investigation, Methodology, Writing—review and editing; Eelco van Anken, Conceptualization, Data curation, Formal analysis, Supervision, Funding acquisition, Validation, Investigation, Visualization, Methodology, Writing—original draft, Project administration, Writing—review and editing

### Author ORCIDs

Andrea Orsi https://orcid.org/0000-0003-2839-1640
Milena Vitale https://orcid.org/0000-0001-7007-402X
Roberto Sitia https://orcid.org/0000-0001-7086-4152
Eelco van Anken http://orcid.org/0000-0001-9529-2701

### Decision letter and Author response

Decision letter https://doi.org/10.7554/eLife.27518.027
Author response https://doi.org/10.7554/eLife.27518.028

## Additional files

### Supplementary files

• Supplementary file 1. Proteomics data and calculations that were used to generate *Figure 6.*
DOI: https://doi.org/10.7554/eLife.27518.023

• Supplementary file 2. List of cell lines, antibodies, and siRNAs used in this study.
DOI: https://doi.org/10.7554/eLife.27518.024

• Transparent reporting form
DOI: https://doi.org/10.7554/eLife.27518.025

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
