## [Decision Letter]

Thank you for submitting your article "The unfolded protein response turns from pro-survival to pro-death to cap the extent of endoplasmic reticulum expansion" for consideration by *eLife*. Your article has been reviewed by three peer reviewers, one of whom, Davis Ng (Reviewer #1), is a member of our Board of Reviewing Editors, and the evaluation has been overseen by Randy Schekman as the Senior Editor. The following individual involved in review of your submission has agreed to reveal their identity: Peter Walter (Reviewer #2).

The reviewers have discussed the reviews with one another and the Reviewing Editor has drafted this decision to help you prepare a revised submission

Summary:

The authors set the case that commonly used agents to induce the UPR, while useful to study signaling, are less desirable for studying the physiological decisions made by the cell to cope with stress due to their pleiotropic effects. The system set up is the controlled expression of the immunoglobulin M heavy chain in HeLa cells. The inspiration for their choice is the established system of B cell development into antibody secreting plasma cells. The process involves the UPR to remodel the ER for differentiation. Here, µs high-level expression causes strong UPR induction that is moderated when the lamda chain is co-expressed consistent with other studies where it is not the volume of the flux but the nature of the protein that is the primary determinant for the extent of UPR induction. The authors then combine quantitative proteomic analysis with ultrastructural measurements of growing ER to their induction profile. Together, these data were used to estimate the changes in steady state protein levels throughout the cell with a focus on ER localized proteins involved in protein homeostasis. Perhaps not surprisingly, BiP and other glucose regulatable chaperones showed the greatest changes over an expanding ER proteome that correlates with ER membrane expansion. The system is well suited the answer the long-standing question of when the UPR switches from being cyto-protective towards being pro-apoptotic.

Although there are specific concerns (see below), the study has many strengths. Of particular interest is the analysis of µs accumulation and how that correlates with the extent of UPR induction. One interesting finding was the following. "Instead, UPR signaling reached maxima when the ratio μ_s_/BiP was highest (Figure 4). These findings support a scenario in which the UPR signals are commensurate with the extent of μ_s_ sequestering the folding machinery, in particular BiP, rather than its accumulation per se." The idea of BiP limitation has been proposed for how the UPR activates, particularly Ire1 but to my knowledge, this is the first time the ratio of a specific stressor has been measured against BiP coordinately with all 3 arms of the UPR. However, the authors should be cautioned that the data in the manuscript do not distinguish the alternate mechanisms so such speculation should be stated as such.

The study is generally well designed and executed. It does an excellent job in validating the protein-based approach to induce ER stress. However, it is in the application of the method where authors encounter problems. In a number of experimental lines, the authors draw conclusions that are not fully supported by the data. In such cases, it is recommended that the text be changed to better reflect the data and any speculation be moved from the Results to the Discussion. In some cases, it might be better to move the data for future publication when the conclusions can be better substantiated. Of broad concern to reviewers is the writing style. There is an overuse of colorful/ornate terms that muddle up the authors' intended points. Because this occurs throughout the manuscript (some examples are outlined below), another round of careful editing is necessary. In the revision, it is also recommended that the text be shortened significantly to improve readability.

The manuscript in the present form is not yet suitable for publication. However, with substantial reorganization of the manuscript and figures, along with clarifications, this study will be of significant interest to the broad audience of *eLife*.

Revisions:

1. Figure 1. The authors state that 50 ng/ml Tm induces "marginal" XBP1 mRNA splicing. Looking at the data, the level of XBP1 mRNA splicing seems to approach 50%, which is by no means modest even when not maximal. Please correct. In addition, it is not accurate to state that a dose of 25 ng/ml Tm "fails" to engender XBP1 mRNA splicing. This is an assumption based on their PCR data, but it should be tested whether or not such low dose induces XBP1 mRNA splicing using specific primers to pick up on spliced from only.

2. In Figure 2, there seems to be a missing a gel crop in the PERK blot between lanes 4-5. In Figure 2, the Ig levels are not "comparable" to plasma cells; they appear to be much higher. Please correct or rephrase.

3. Figure 3. The authors overstate how their results distinguish flux of proteins entering the ER from simple accumulation of unfolded proteins in the ER lumen. Flux is not measured directly in this experiment, and an alternative explanation for the data is that IgL simply competes with BiP for binding sites on μ_s_, resulting in an apparent "rescue" that results from the freeing of BiP that would otherwise be bound to μ_s_.

4. Figure 4. The rise in BiP levels over those of Ig heavy chain is marginal. The authors should include metrics for statistical significance for each data point.

5. Figure 6. The authors should include quantification next to the corresponding pie chart slices for other organelles.

6. The conclusion that BiP behaves essentially as the ER "fuel gauge" enabling the cell to continue to expand the ER and attempt homeostasis is well supported by the data. However, there are no data to dismiss that direct binding of unfolded Ig heavy chain peptides could not engage the ER stress sensors as well. It could be possible that by flooding the ER with a single type of client, UPR activation results from the concomitant actions of the single ER client serving as BiP sink and unfolded peptides engaging the ER stress sensors. Moreover, it could be possible that the direct binding model could also help in fine-tuning the response. The data in this paper in no way refute the model that ER sensors are activated directly by unfolded protein binding.

7. Why is there no increase in BiP levels in the positive control (far left lane)?

8. The authors claim that Ire1a was replaced with Ire1a-GFP in cells. I don't see any description of CRISPR tagging, use of IRE1 KO cells or otherwise. This needs to be fully described.

9. How long were cells induced with Mif? While growing in the plates, only prior to placing on plates? This is not clear from the figure legend.

10. Figure 4 is a misrepresentation of the data. The data are not collected continuously. The time line and shading suggest continuous sampling. A standard plot with time points is the proper way to display these data.

11. Figure 5 lower panels. The dark blue histograms are often difficult to see against the dark EM background. Please use a better color or make a distinct histogram figure. Separating the histograms is not a great way to compare the data. The histograms should be presented side by side. This is true for 8F and 9E, too.

12. Figure 6. The choice of two dark blues for the pie charts is confusing as it inadvertently suggests the total protein increase is BiP. Pick a different color for the ER in the total proteome.

Since lumenal J proteins are key BiP co-chaperones and known to increase robustly with ER stress, it would make sense to include these in the analyses.

13. Abstract: "the UPR transitions from acute to full-geared activation…". The terms "acute" and "full-geared" are confusing in this context. Aren't they the same? Acute is generally meant to mean severe.

14. The authors may wish to restate that the µs induced HeLa cells "triggered a full-blown UPR" (page 5 bottom) for a number of reasons. First, the time of induction with tunicamycin or thapsigargin is 1.5 h versus 16 h for µs expression. They really aren't directly comparable. Second, the extent of induction is different. The extent of XBP1 splicing and ATF6 processing is less in the µs lanes. There is more BiP protein at steady state in the µs expressed cells but that's misleading because that's 16 h of induced expression versus 1.5 h and BiP is a very stable protein. Third, if µs expression did lead to "full-blown" UPR activation (vague, but defined in the text as Tm/Tg treatment), I see that as problematic because the authors have argued persuasively that such treatment is non-physiological and toxic. Based on that argument, I would expect that ER stress that is manageable would cause a more modest level of induction, which is exactly what is observed.

15. The passage: "Thus, the full-blown UPR that was elicited by μ_s_ expression was well tolerated by the cells, unlike prolonged treatment with Tm at concentrations that triggered a full-blown UPR (Figure 1)." Is tolerance due to UPR induction? Has this experiment been performed in cell lines deficient in any of the 3 UPR branches?

16. The passage: "Since the cells that inducibly overexpress μ_s_ showed maximal UPR activation with no major negative impact on cell growth-unlike cells treated with UPR eliciting drugs-we surmised that the μ_s_ expressing cells underwent successful homeostatic readjustment of the ER." This statement seems to contradict the stated basis for setting up this system, that the drugs are toxic, not because they simply induce the UPR, but that they disrupt the function of many proteins.

[Editors' note: further revisions were requested prior to acceptance, as described below.]

Thank you for resubmitting your work entitled "The unfolded protein response turns from pro-survival to pro-death to cap the extent of endoplasmic reticulum expansion" for further consideration at *eLife*. Your revised article has been favorably evaluated by Randy Schekman (Senior editor), a Reviewing editor, and two reviewers.

Although the manuscript is improved from the original version, there are claims that reviewers view as not fully supported. While the consensus view remains that the µs overexpression system of ER stress is well designed and characterised, the use of the system to understand specific questions of UPR physiology are not as well designed. Because the study is extensive, it is the advice of the reviewers to restate the claims as instructed and/or remove data that can be published elsewhere with additional work. It is our view that the manuscript can be published after some extensive reorganisation of the data and major text changes.

[Editors’ note: the following was also passed to the authors via correspondence with the editorial staff, further an author query.]

Regarding splitting the paper into two, it is the consensus view of the reviewers that the innovation of establishing a more physiological form of inducing ER stress and its validation to be the strength of the manuscript. That should be the focus of paper one. For paper two, we would caution the authors to consider very carefully how to proceed. We believe this is where the authors wish to use the system to answer specific questions regarding how the UPR responds to and alleviates ER stress. While the goal is important and an excellent use of the system, it is clear that the authors tried to do too much here, which made the supporting data incomplete at this point. Unless the authors have expanded this part of the study significantly and completely addressed reviewer concerns, it is not recommended that this part to be co-submitted. It should be submitted separately at another time.

[Editors' note: following the editors’ advice, the authors resubmitted and further revisions were requested prior to acceptance, as described below.]

Thank you for resubmitting your work now entitled "Ratiometric sensing of BiP-client versus BiP levels by the unfolded protein response determines its signaling amplitude" for further consideration at eLife. Your revised article has been favorably evaluated by Randy Schekman (Senior editor), a Reviewing editor, and one reviewer.

The manuscript has been improved but there is one minor, but significant, issue that needs to be addressed before acceptance, as outlined below:

*Reviewer #3:*

The heavily revised manuscript is dramatically improved and I will be interested to see the companion manuscript. The wording now agrees well with the presented data and I consider the manuscript acceptable with one small correction.

I remain adamant that Figure 4 is not appropriate as presented. The data in 4C are not continuous. They are discrete. I have no problem with the authors presenting their data as a heat map. That is fine. However, it should be clear that each data point is a defined block representing a specific time point. The data, as currently plotted, in a smeared form indicate that there are multiple data points that blur together, as if data were sampled every few minutes. Just present the heat map as blocks instead of a smear and I accept the revised manuscript.

The request from the reviewer does not involve any additional experimentation but modification in presentation of the data.

[Editors’ note: the original Decision letter and Author response included references to data which have subsequently been omitted and are therefore not discussed in this edited version.]

---

## [Author Response]

We were of course delighted to read that the reviewers found that “there are many strengths to the study”, that “the study is generally well designed and executed”, and that “this study will be of significant interest to the broad audience of eLife”. As you may recall, in the paper we present a HeLa cell model for endoplasmic reticulum (ER) stress driven by a proteostatic insult in the form of the heavy chain of IgM (µs). Since the cells can cope with the insult, we created a system that the reviewers agree “is well suited to answer the long-standing question of when the UPR switches from being cyto-protective towards being pro-apoptotic”.

Your letter moreover highlighted that “one interesting finding was the following: ‘Instead, UPR signaling reached maxima when the ratio μ_s_/BiP was highest (Figure 4). These findings support a scenario in which the UPR signals are commensurate with the extent of μ_s_ sequestering the folding machinery, in particular BiP, rather than its accumulation per se.’ (Results). The idea of BiP limitation has been proposed for how the UPR activates, particularly Ire1 but to my knowledge, this is the first time the ratio of a specific stressor has been measured against BiP coordinately with all 3 arms of the UPR”, and that our study”does an excellent job in validating the protein-based approach to induce ER stress”. We include point-by-point answers to the reviewers’ questions below.

[…] Revisions:1. Figure 1. The authors state that 50 ng/ml Tm induces "marginal" XBP1 mRNA splicing. Looking at the data, the level of XBP1 mRNA splicing seems to approach 50%, which is by no means modest even when not maximal. Please correct. In addition, it is not accurate to state that a dose of 25 ng/ml Tm "fails" to engender XBP1 mRNA splicing. This is an assumption based on their PCR data, but it should be tested whether or not such low dose induces XBP1 mRNA splicing using specific primers to pick up on spliced from only.

In view of the reviewers’ considerations we adapted the text.

2. In Figure 2, there seems to be a missing a gel crop in the PERK blot between lanes 4-5. In Figure 2, the Ig levels are not "comparable" to plasma cells; they appear to be much higher. Please correct or rephrase.

We thank the reviewers for their concern, but lanes 4 and 5 were adjacent already in the original PERK blot, so no separator line is necessary.

In Figure 2, the Ig levels are not "comparable" to plasma cells; they appear to be much higher. Please correct or rephrase.

In view of the reviewers’ considerations we adapted the text.

3. Figure 3. The authors overstate how their results distinguish flux of proteins entering the ER from simple accumulation of unfolded proteins in the ER lumen. Flux is not measured directly in this experiment, and…

In view of the reviewers’ considerations we adapted the text.

[…] an alternative explanation for the data is that IgL simply competes with BiP for binding sites on μ_s_, resulting in an apparent "rescue" that results from the freeing of BiP that would otherwise be bound to μ_s_.

BiP indeed acts as a “place-holder” for 1 and tightly binds to µ_s_ (exclusively at the C_H_1 domain where 1 will bind), which we refer to in the text (and further highlight in a schematic figure). We adapted the text to make this point slightly clearer.

4. Figure 4. The rise in BiP levels over those of Ig heavy chain is marginal. The authors should include metrics for statistical significance for each data point.

We have performed the statistical analysis. Evidently, before the onset of µ_s_ expression, the excess of BiP over µ_s_ is statistically significant. Between 2-8 hours there is no statistically significant difference between BiP and µ_s_ levels. Then, between 8-12 hrs, µ_s_ exceeds BiP levels. The statistical significance is marginal, but in this time span also UPR output levels are highest. Between 16-32 hrs again there is no statistical difference between µ_s_ and BiP levels, whereupon BiP levels exceed µ_s_ levels in a highly significant manner in the chronic phase.

To report on the significance of the differences in expression levels in an intuitive manner (see also the response below to minor point 18), we thought to display a time bar with a color-coding scale, ranging from dark blue (BiP > µ_s_; p < 0.001) to purple (BiP ≈ µ_s_) to red (µ_s_ > BiP; p < 0.05).

Since we do not think that the method is accurate enough to establish exactly by what margin µ_s_ eclipses BiP levels, we want to emphasize the opposite of which we are very confident, namely, that BiP levels transiently *do not* eclipse those of µ_s_. We therefore feel that the accompanying sentence in the text still best expresses how the data should be interpreted: “Early upon the onset of µ_s_ expression and before BiP induction was fully underway, however, µ_s_ levels transiently were at a 1:1 stoichiometry with those of BiP, or possibly, μ_s_ levels even slightly exceeded BiP levels (Figure 4).”

5. Figure 6. The authors should include quantification next to the corresponding pie chart slices for other organelles.

The way the calculations of the data and the representation in this figure were designed was to emphasize that µ_s_ expression causes few if any changes in the proteome except that the ER increases considerably in protein mass in comparison with the rest of the cell. While we believe the approach we have chosen was ideal for demonstrating just that, we would be hesitant to give too much credence to the absolute values we obtained per organelle. To display them next to the graphs would distract from the message and suggest an accuracy that we cannot vouch for. Besides the real quantifications are already in Supplementary file 1 and easily traceable for who might be interested in those details.

6. The conclusion that BiP behaves essentially as the ER "fuel gauge" enabling the cell to continue to expand the ER and attempt homeostasis is well supported by the data. However, there are no data to dismiss that direct binding of unfolded Ig heavy chain peptides could not engage the ER stress sensors as well. It could be possible that by flooding the ER with a single type of client, UPR activation results from the concomitant actions of the single ER client serving as BiP sink and unfolded peptides engaging the ER stress sensors. Moreover, it could be possible that the direct binding model could also help in fine-tuning the response. The data in this paper in no way refute the model that ER sensors are activated directly by unfolded protein binding.

We reconsidered this issue and concluded that the models of UPR sensor activation are only in apparent contradiction, but physiologically interchangeable and therefore easy to reconcile. We adapted the Discussion to make this point clear.

7. Why is there no increase in BiP levels in the positive control (far left lane)?

We assume the reviewers allude to the BiP levels upon Tm treatment in Figure 2. Since cells were treated for only 4 hrs with 5 µg/ml Tm no increase of BiP is detectable at the protein level – it would require a longer Tm treatment (12 hrs) to perceive changes for BiP at the protein level, but at that stage cells are already quite sick from the Tm treatment.

8. The authors claim that Ire1a was replaced with Ire1a-GFP in cells. I don't see any description of CRISPR tagging, use of IRE1 KO cells or otherwise. This needs to be fully described.

The reconstitution is amply described in the Material and methods section of the manuscript and the validation of the reconstitution is shown in the figure supplement to Figure 2. Since the tool we built is almost identical to that built by Han Li (PNAS, 2010), we felt it would be superfluous to present it as a result, which is why we “relegated” the description of the knock-out and reconstitution to the Material and methods, but if the reviewers insist, we will promote them to the Results section.

9. How long were cells induced with Mif? While growing in the plates, only prior to placing on plates? This is not clear from the figure legend.

The cells were treated for 7 days with mifepristone, starting from when cells attached (usually 8 hrs) after plating. We now state that in the figure legend and describe the procedure better in the Material and methods section.

10. Figure 4 is a misrepresentation of the data. The data are not collected continuously. The time line and shading suggest continuous sampling. A standard plot with time points is the proper way to display these data.

We now provide standard plots for each UPR output that we have assessed in the Figure 4—figure supplement 1–Figure 4—figure supplement 3. We do understand the criticism that the color-based heat maps would suggest continuous sampling, but since we provide one example (Figure 4) and now also the plots summarizing the data for that and repeat experiments on which the heat maps are based as supplements, we believe that the reader will not be led astray in thinking that there was continuous sampling.

We believe that the color-based heat maps, in particular in combination with the color-based heat map we now produced to depict statistical significance of differences in expression levels between BiP and µ_s_ (see response above to minor point 4), give an easy to grasp and intuitive visual summary of the message we wish to convey, namely that UPR signaling output levels are commensurate with BiP levels running short relative to µ_s_ levels.

11. Figure 5 lower panels. The dark blue histograms are often difficult to see against the dark EM background. Please use a better color or make a distinct histogram figure. Separating the histograms is not a great way to compare the data. The histograms should be presented side by side. This is true for 8F and 9E, too.

We adapted the figures following the reviewers’ suggestions.

12. Figure 6. The choice of two dark blues for the pie charts is confusing as it inadvertently suggests the total protein increase is BiP. Pick a different color for the ER in the total proteome.

We adapted the figures following the reviewers’ suggestions.

Since lumenal J proteins are key BiP co-chaperones and known to increase robustly with ER stress, it would make sense to include these in the analyses.

Reports on the robust upregulation of ER lumenal J (ERdj) proteins are all based on fold changes of transcripts. The tendency in -omics studies to focus on fold changes leads to a particular emphasis in the interpretation of biological phenomena, which sometimes even may obscure other biologically meaningful aspects. In our study we have chosen to estimate protein levels in ppm of the total proteome, which gives another perspective on what are robust increases.

For instance, according to our proteomics data, PDI is upregulated “only” 1.6 fold, but since PDI is an abundant protein at basal conditions, being already 4.5x10^4^ ppm, which equates to approximately 2x10^7^ copies per non-stressed cell, it implies that more than 1x10^7^ extra molecules of PDI are present after 7 days of µ_s_ expression. Ten million extra copies of PDI seems to us a rather robust change, despite the low fold change!

From the data presented in Supplementary file 1 it is evident that four ERdj proteins were so lowly expressed or not expressed at all in our model cell line that we did not detect them by proteomics: ERdj1 (DNAJC1), ERdj2 (SEC63), ERdj4 (DNAJB9), and ERdj7 (DNAJC25), while ERdj5 (DNAJC10) was barely detectable by proteomics. ERdj3 (DNAJB11) and ERdj6 (DNAJC3) were well-detectable by proteomics, but still in low amounts. Our proteomics survey estimated ERdj3 and ERdj6 to go up about 3-6 fold from about 2-6x10^2^ ppm to 2x10^3^ ppm, which would amount to 1x10^6^ copies per cell for either ERdj protein after 7 days of µ_s_ expression; in other words there are about 1-2x10^6^ total ERdj protein extra copies per cell, which arguably is a less robust change than that of PDI, even though at the mRNA level the fold change may be much more flagrant than for PDI. For comparison: we estimated BiP levels after 7 days of µ_s_ expression to be about 2x10^8^ copies per cell. Thus, altogether it seems that BiP levels exceed those of its co-chaperones about a hundred fold. From these estimates it thus appears that either a single ERdj protein can cater 100 BiP proteins simultaneously or that a majority of BiP molecules operates without co-chaperoning.

Since our aim was to obtain an idea of how the cell, and, in particular the ER, changes in terms of protein (mass) content, we chose the “cut off” for in-depth analysis such that it pertained only to the proteins that were among the 500-600 most abundant proteins, as explained in the manuscript. The ERdjs did not make that cut and if we would display them in the way we display our data in the proteomics figure, the ERdjs would be still invisible, as the most lowly abundant protein within this top 500-600, TRAPd, is already barely visible in the pie diagram.

We therefore do not feel we should add data on ERdjs to the figure. For who would be interested in the ERdjs, their expression data are readily available in Supplementary file 1 did however decide to briefly touch in the results on the finding that BiP vastly outnumbers its co-chaperones, the ERdjs. We moreover acknowledge better the current knowledge on ERdjs in the introduction by citing a recent review by Melnyk et al.

13. Abstract: "the UPR transitions from acute to full-geared activation…". The terms "acute" and "full-geared" are confusing in this context. Aren't they the same? Acute is generally meant to mean severe.

To avoid confusion we corrected the abstract such that there is now a comma between acute and full-geared (which lacked before): “acute, full-geared activation”.

14. The authors may wish to restate that the µs induced HeLa cells "triggered a full-blown UPR" (page 5 bottom) for a number of reasons. First, the time of induction with tunicamycin or thapsigargin is 1.5 h versus 16 h for µs expression. They really aren't directly comparable. Second, the extent of induction is different. The extent of XBP1 splicing and ATF6 processing is less in the µs lanes. There is more BiP protein at steady state in the µs expressed cells but that's misleading because that's 16 h of induced expression versus 1.5 h and BiP is a very stable protein. Third, if µs expression did lead to "full-blown" UPR activation (vague, but defined in the text as Tm/Tg treatment), I see that as problematic because the authors have argued persuasively that such treatment is non-physiological and toxic. Based on that argument, I would expect that ER stress that is manageable would cause a more modest level of induction, which is exactly what is observed.15. The passage: "Thus, the full-blown UPR that was elicited by μ_s_ expression was well tolerated by the cells, unlike prolonged treatment with Tm at concentrations that triggered a full-blown UPR (Figure 1)." Is tolerance due to UPR induction? Has this experiment been performed in cell lines deficient in any of the 3 UPR branches?16. The passage: "Since the cells that inducibly overexpress μ_s_ showed maximal UPR activation with no major negative impact on cell growth-unlike cells treated with UPR eliciting drugs-we surmised that the μ_s_ expressing cells underwent successful homeostatic readjustment of the ER." This statement seems to contradict the stated basis for setting up this system, that the drugs are toxic, not because they simply induce the UPR, but that they disrupt the function of many proteins.

The above three points that the reviewers raise are a bit bewildering to us as they seem to suggest (contrary to the earlier points that were raised) that the reviewers do not see what the merits of our model using a proteostatic insult are with respect to “traditional” ER stress eliciting drugs.

In short: Tm etc. may trigger full-blown UPR output levels, i.e. maximally obtainable levels of splicing, ATF6a cleavage and CHOP expression, but ER homeostatic readjustment cannot be achieved, since the cells succumb to pleiotropic effects before. From the time course experiment that we show (Figure 4) it is evident that inducible µ_s_ expression leads to similar full-blown UPR output levels (albeit that maximal splicing levels are reached at a slower pace), but the cells do not suffer from pleiotropic effects and, thus, ER homeostatic readjustment can be achieved. As such, we were in a position to demonstrate that the UPR is essential for this ER homeostatic readjustment (Figure 7).

Thus, full-blown UPR output levels are not detrimental to the cells per se. It is the pleiotropic effects of the ER stress eliciting drugs that kills them. Cells die from Tm treatment likely due to glycoproteins being depleted at destinations where they have to exert their function, and similar reasoning would explain the toxic effects of DTT or Tg. The UPR to some extent mitigates these problems and thereby even prolongs the life of cells treated with Tm etc. Accordingly, we found that cells in which UPR pathways are ablated succumb at even lower doses of Tm than WT cells. This finding demonstrates that strategies to investigate potential pro-apoptotic roles of the UPR in the context of cells treated with ER-stress-eliciting drugs by default are in vain.

Since we apparently did not make this notion clear enough to the reviewers, as it seemed from minor points 24-26, we decided to include this finding in Figure 1 to highlight better why strategies to elucidate how the UPR can transition from a cyto-protective to pro-apoptotic output are in vain when ER-stress eliciting drugs are used. We adapted the text accordingly and moved the original Figure 7—figure supplement 1 to become Figure 1—figure supplement 1.

[Editors' note: further revisions were requested prior to acceptance, as described below.]

Although the manuscript is improved from the original version, there are claims that reviewers view as not fully supported. While the consensus view remains that the µs overexpression system of ER stress is well designed and characterised, the use of the system to understand specific questions of UPR physiology are not as well designed. Because the study is extensive, it is the advice of the reviewers to restate the claims as instructed and/or remove data that can be published elsewhere with additional work. It is our view that the manuscript can be published after some extensive reorganisation of the data and major text changes.

We were delighted to hear that our revised work entitled "The unfolded protein response turns from pro-survival to pro-death to cap the extent of endoplasmic reticulum expansion" has been favorably evaluated.

In spite of the positive evaluation, your letter stated that "there are claims that reviewers view as not fully supported", and the reviewers advised us "to restate the claims as instructed and/or remove data that can be published elsewhere with additional work", while the view at *eLife* was that "the manuscript can be published after some extensive reorganization of the data and major text changes."

If we follow the reviewers' advice, we simply remove the parts to which they have raised objections, and they still would like to see it published. If the editors would agree with this option, it would be ideal for us, since we are anxious too to get this work out. Yet, the second part contains almost all issues that have been central to the reviewing history of our manuscript so far at *eLife*, such that if we remove the second part from this submission, it would make sense to resubmit that second part as a separate manuscript to *eLife* again, since it would have been extensively reviewed at *eLife* already. We moreover think that the study becomes really too extensive for a single paper, in particular once extra data would be included if according to the reviewers they would be sufficient to satisfyingly support our claims.

[Editors' note: further revisions were requested prior to acceptance, as described below.]

Reviewer #3:The heavily revised manuscript is dramatically improved and I will be interested to see the companion manuscript. The wording now agrees well with the presented data and I consider the manuscript acceptable with one small correction.I remain adamant that Figure 4 is not appropriate as presented. The data in 4C are not continuous. They are discrete. I have no problem with the authors presenting their data as a heat map. That is fine. However, it should be clear that each data point is a defined block representing a specific time point. The data, as currently plotted, in a smeared form indicate that there are multiple data points that blur together, as if data were sampled every few minutes. Just present the heat map as blocks instead of a smear and I accept the revised manuscript.The request from the reviewer does not involve any additional experimentation but modification in presentation of the data.

We adapted the figure as requested.